# Mitochondrial MICOS complex genes, implicated in hypoplastic left heart syndrome, maintain cardiac contractility and actomyosin integrity

Katja Birker[1†], Shuchao Ge[1†], Natalie J Kirkland[2†], Jeanne L Theis[3†], James Marchant[1], Zachary C Fogarty[4], Maria A Missinato[1], Sreehari Kalvakuri[1], Paul Grossfeld[5], Adam J Engler[2], Karen Ocorr[1], Timothy J Nelson[6], Alexandre R Colas[1], Timothy M Olson[7], Georg Vogler[1*], Rolf Bodmer[1*]

[1]Development, Aging and Regeneration Program, Center for Genetic Disorders & Aging Research, Sanford Burnham Prebys Medical Discovery Institute, San Diego, United States; [2]Department of Bioengineering, Sanford Consortium for Regenerative Medicine, UCSD, School of Medicine, San Diego, United States; [3]Cardiovascular Genetics Research Laboratory, Mayo Clinic, Rochester, United States; [4]Division of Computational Biology, Department of Quantitative Health Sciences, Mayo Clinic, Rochester, United States; [5]Department of Pediatrics, UCSD School of Medicine, La Jolla, Rady's Hospital MC 5004, San Diego, United States; [6]Center for Regenerative Medicine, Division of Pediatric Cardiology, Department of Pediatric and Adolescent Medicine, Division of General Internal Medicine, Department of Molecular and Pharmacology and Experimental Therapeutics, Mayo Clinic, Rochester, United States; [7]Department of Cardiovascular Medicine, Division of Pediatric Cardiology, Department of Pediatric & Adolescent Medicine, Cardiovascular Genetics Research Laboratory, Mayo Clinic, Rochester, United States

*For correspondence:
gvogler@sbpdiscovery.org (GV);
rolf@sbpdiscovery.org (RB)

†These authors contributed equally to this work

Competing interest: The authors declare that no competing interests exist.

**Abstract** Hypoplastic left heart syndrome (HLHS) is a severe congenital heart disease (CHD) with a likely oligogenic etiology, but our understanding of the genetic complexities and pathogenic mechanisms leading to HLHS is limited. We performed whole genome sequencing (WGS) on 183 HLHS patient-parent trios to identify candidate genes, which were functionally tested in the *Drosophila* heart model. Bioinformatic analysis of WGS data from an index family of a HLHS proband born to consanguineous parents prioritized 9 candidate genes with rare, predicted damaging homozygous variants. Of them, cardiac-specific knockdown (KD) of mitochondrial MICOS complex subunit *dCHCHD3/6* resulted in drastically compromised heart contractility, diminished levels of sarcomeric actin and myosin, reduced cardiac ATP levels, and mitochondrial fission-fusion defects. These defects were similar to those inflicted by cardiac KD of ATP synthase subunits of the electron transport chain (ETC), consistent with the MICOS complex's role in maintaining cristae morphology and ETC assembly. Five additional HLHS probands harbored rare, predicted damaging variants in *CHCHD3* or *CHCHD6*. Hypothesizing an oligogenic basis for HLHS, we tested 60 additional prioritized candidate genes from these patients for genetic interactions with *CHCHD3/6* in sensitized fly hearts. Moderate KD of *CHCHD3/6* in combination with *Cdk12* (activator of RNA polymerase II), *RNF149* (*goliath*, E3 ubiquitin ligase), or *SPTBN1* (*β-Spectrin*, scaffolding protein) caused synergistic heart defects, suggesting the likely involvement of diverse pathways in HLHS. Further elucidation

of novel candidate genes and genetic interactions of potentially disease-contributing pathways is expected to lead to a better understanding of HLHS and other CHDs.

## Editor's evaluation

In the revised version, Birker et al. add experiments in flies and human iPSC-derived cardiomyocytes and add new text that further supports their central claim that mutations in MICOS complex components mediate HLHS by reducing sarcomere integrity and heart contractility. The ability to go from identifying variants by whole genome sequencing in HLHS patients to generating oligogenic animal models to test whether these variants produce cardiac phenotypes is well demonstrated here and highlights the importance of model organisms in disease research. Overall, the manuscript is improved, and the data support the claims.

## Introduction

Hypoplastic left heart syndrome (HLHS) is a birth defect that accounts for 2–4% of congenital heart defects (CHDs), equal to 1000–2000 HLHS births in the United States per year. HLHS has been proposed to be caused by genetic, epigenetic, or environmental factors (*Crucean et al., 2017*; *Liu et al., 2017*; *Yagi et al., 2018*; *Grossfeld et al., 2019*). The severe cardiac characteristics of HLHS include aortic and mitral stenosis or atresia, and reduced size of the left ventricle and aorta; however, there is a spectrum of cardiac phenotypes that can underly HLHS pathophysiology (*Theis et al., 2015a*; *Crucean et al., 2017*; *Mussa and Barron, 2017*; *Grossfeld et al., 2019*). If not treated with reconstructive heart surgeries or cardiac transplantation, infants born with HLHS will not survive (*Grossfeld et al., 2019*). To date, the standard treatment for this disease is a three-stage surgical procedure, which begins neonatally and aims overall to achieve right ventricle-dependent systemic circulation and deliver oxygen-poor blood more directly to the lungs (*Mussa and Barron, 2017*). Although the surgical procedures correctly divert left ventricular function to the right ventricle, there is a subgroup of HLHS patients who are at risk of latent heart failure, which is often preceded by reduced ejection fraction (*Altmann et al., 2000*; *McBride et al., 2008*; *Theis et al., 2015b*).

Although several studies have examined the molecular underpinnings of HLHS, the number of genes associated with this disease is small (e.g. *NKX2-5, NOTCH1, ETS1, MYH6,* and *LRP2,* and *CELSR1*), and they are not yet conclusively determined as causal for HLHS (*Garg et al., 2005*; *Ye et al., 2010*; *Kobayashi et al., 2014*; *Theis et al., 2015a*; *Tomita-Mitchell et al., 2016*; *Theis et al., 2020*; *Theis, 2021*; *Theis et al., 2022*). Defining pathogenic mechanisms has proved elusive given the oligogenic complexity of HLHS. Overall, there is a great need to functionally evaluate newly emerging HLHS candidate genes to understand how they may contribute to the molecular, cellular, and morphological processes underlying HLHS.

*Drosophila* is well-suited for modeling genetic underpinnings of CHDs: many of the genes and gene programs found in the *Drosophila* heart are evolutionarily conserved, including a core set of cardiogenic transcription factors and inductive factors (e.g. *Nkx2-5/tinman*) (*Bodmer, 1995*; *Cripps and Olson, 2002*; *Bier and Bodmer, 2004*; *Bodmer and Frasch, 2010*; *Ahmad, 2017*), approximately 75% of known human disease-causing genes having fly orthologs (*Bodmer and Frasch, 2010*; *Pandey and Nichols, 2011*; *Ugur et al., 2016*), and the developing mammalian and *Drosophila* hearts share developmental similarities, such as their origin within the mesoderm.

Mitochondria have been postulated to play a critical role in HLHS pathogenesis. For example, a recent study reported that cardiomyocytes derived from iPSCs of HLHS patients (iPSC-CM), who later developed right ventricular failure, had reduced mitochondrial concentration, ATP production, and contractile force (*Paige et al., 2020*). This study revealed downregulated expression of genes involved in mitochondrial processes, such as ATP synthesis coupled electron transport. Another study of HLHS patient-derived iPSC-CMs revealed reduced mitochondrial size, number, and malformed mitochondrial inner membranes using transmission electron microscopy (*Yang et al., 2017*). Similarly, an HLHS mouse model with *Sap130* and *Pcdha9* mutations showed mitochondrial defects manifested as reduced cristae density and smaller mitochondrial size (*Liu et al., 2017*). Despite a lack of

understanding of the exact mitochondrial mechanisms underlying HLHS pathogenesis, recent experimental and bioinformatic data suggest an underlying role of mitochondria in HLHS.

Here, a cohort of 183 HLHS proband-parent trios underwent whole genome sequencing (WGS) to identify candidate genes, including a prioritized consanguineous family where genes harboring rare, predicted damaging homozygous variants were investigated (*Theis and Olson, 2022*). Among the resulting candidate HLHS genes tested in *Drosophila*, cardiac-specific knockdown (KD) of *Chchd3/6* (*coiled-coil-helix-coiled-coil-helix-domain-containing protein* 6) of the MICOS (mitochondrial contact site and cristae organization system) complex exhibited severe heart structure and function defects. The MICOS complex is an eight-subunit complex in mammals (five in *Drosophila*) located in the inner mitochondrial membrane that is necessary to maintain cristae morphology and ATP production. It is closely associated and interacts with SAMM50 (sorting and assembly machinery, CG7639), which is located in the outer mitochondrial membrane (*Ott et al., 2012*; *Kozjak-Pavlovic, 2017*). The MICOS complex's role in cardiac development and functional homeostasis is not known but is likely important for efficient ATP production. We observed reduced contractility upon cardiac-specific *Chchd3/6* KD, diminished sarcomeric Actin and Myosin levels, as well as severe mitochondrial morphology defects, which manifested as fragmented and aggregated structures. Similar phenotypes were observed upon cardiac KD of other MICOS complex genes, as well as other mitochondrial genes such as ATP synthase (complex V), specifically ATP synthase B and β. We also found significantly diminished proliferation of human induced pluripotent stem cell (iPSC)-derived ventricular-like cardiomyocytes (VCMs) upon KD of MICOS genes. Finally, a family-based candidate gene interaction screen in *Drosophila* revealed three genes that genetically interact with *Chchd3/6*: *Cdk12* (activator RNA polymerase II activator), *RNF149* (*goliath, gol,* E3 ubiquitin ligase), *SPTBN1* (*β Spectrin, β-Spec,* scaffolding protein). In summary, *Chchd3/6* and other components important for mitochondrial homeostasis were identified as critical for establishing and maintaining cardiac structure and function, and likely contribute to HLHS and/or latent heart failure following surgical palliation.

## Results

### Family phenotype

Family 11 H is of white ancestry and comprised of a male with HLHS, his parents, and two siblings, they were all phenotypically characterized by echocardiography and underwent WGS. A homozygous recessive disease mode of inheritance was postulated due to reported consanguinity between the mother and father and absence of structural and myopathic heart disease in the parents (*Figure 1A*). The siblings also had normal echocardiograms. The 11 H proband had latent decline of right ventricular ejection fraction several years after surgical palliation. In addition to HLHS, he was diagnosed with developmental delay, cerebral and cerebellar atrophy, white matter loss, decreased muscle mass, and a body mass index <1%, traits that have previously been related to mitochondrial dysfunction (*Alston et al., 2017*; *Romanello and Sandri, 2016*).

### Whole genome sequencing and bioinformatics analysis of 11H family

Array comparative genomic hybridization ruled out a chromosomal deletion or duplication in the proband. WGS was carried out on genomic DNA samples from the five family members, based on paired-end reads that passed quality control standards; 99.4% of the reads mapped to the genome. After marking and filtering out duplicate reads, over 91% of the hg38 human reference genome had coverage. The average depth across the genome was 63 X and an average of 89% of the genome demonstrated a minimal read depth of 20 reads. Filtering for rare variants that were homozygous in the HLHS proband revealed nine candidate genes. Three genes had a missense variant (*SZT2, MTRR, MBTPS1*) whereas the remaining six genes were found to have a non-coding variant within the promoter (*CAP1, DGKE*), 5′ untranslated region (*RHBDL2, RNF149, C17orf67*), or intron (*CHCHD6*) (*Marian et al., 2011*). While six of the variants were also found to be homozygous in an unaffected sibling, the associated candidate genes were not excluded from downstream analyses based on the postulated oligogenic nature of HLHS, and incomplete penetrance of individual variants, as observed in a digenic mouse model (*Yagi et al., 2018*).

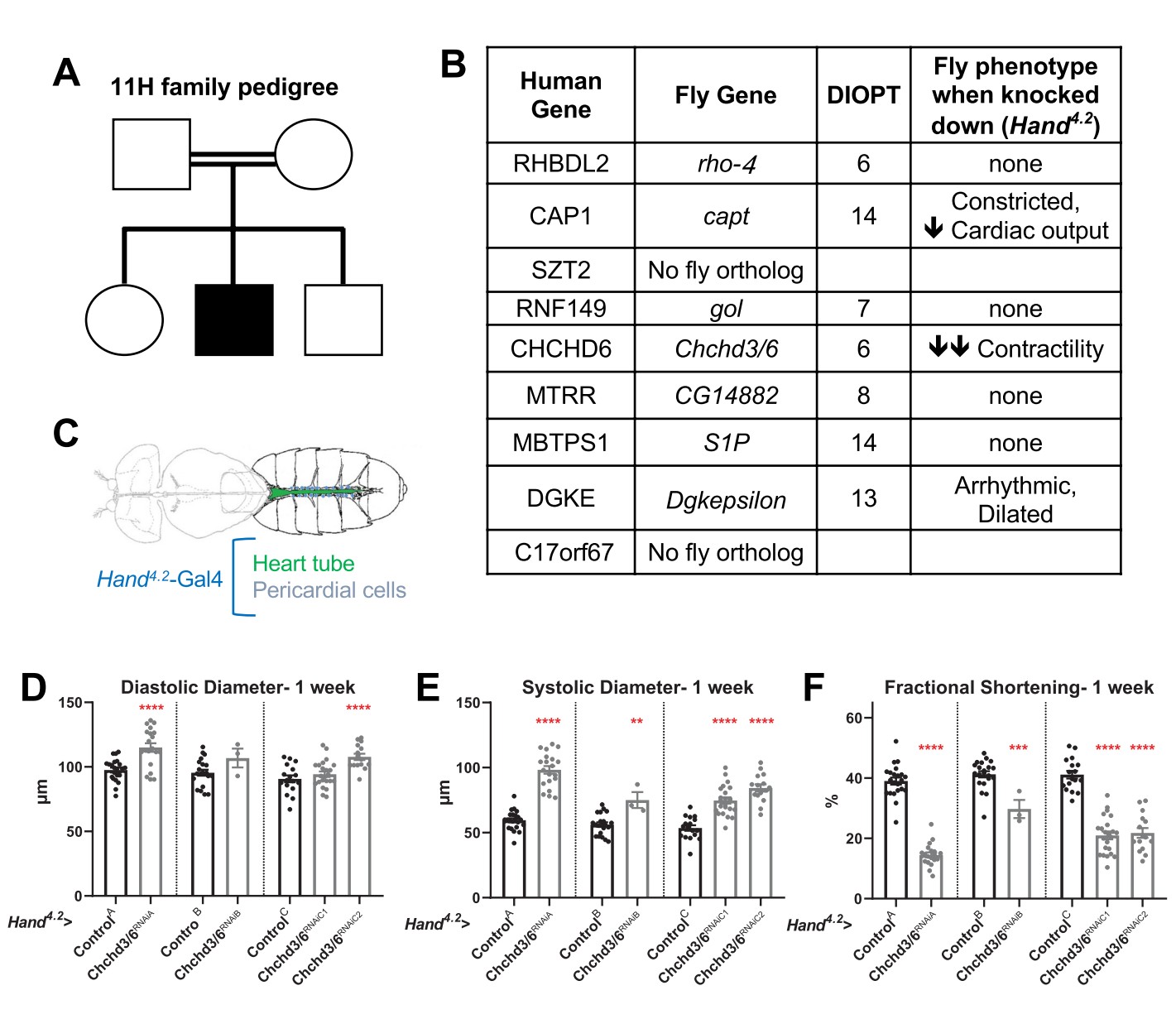

**Figure 1.** Prioritization of *CHCHD6* in HLHS proband and its *Drosophila* ortholog *Chchd3/6*. (**A**) Pedigree of index family 11 H. The family includes consanguineous parents (denoted by double horizontal lines) without cardiac defects, one son with HLHS (proband), and two siblings without cardiac defects. (**B**) List of 9 candidate genes derived from proband 11 H with corresponding *Drosophila* orthologs. Orthology based on DIOPT score. Conserved *Drosophila* candidate HLHS genes were knocked down individually in the *Drosophila* heart using the *Hand*[4.2]*-Gal4* driver. The functional phenotypes listed were significantly different relative to Control[A] or Control[B] and were measured in 1-week-old female *Drosophila* hearts. (**C**) Schematic of *Drosophila* highlighting the abdominal region which includes the heart tube and flanking pericardial cells, where the *Hand*[4.2]-Gal4 driver is expressed. Image adapted from Figure 1A of *Xie et al., 2013*. (**D**) End-Diastolic diameter (EDD), (**E**) End-systolic diameter (ESD), and (**F**) fractional shortening (FS) from 1-week-old female *Hand*[4.2]*-Gal4>Chchd3/6* flies.

The online version of this article includes the following source data and figure supplement(s) for figure 1:

**Source data 1.** List of 9 candidate genes derived from proband 11 H with corresponding *Drosophila* orthologs.

**Figure supplement 1.** *Chchd3/6* KD in the *Drosophila* heart causes reduced fractional shortening due to systolic dysfunction.

## Candidate HLHS gene knockdown in *Drosophila* reveals requirement for *Chchd3/6* in establishing cardiac structure and function

To test whether the HLHS candidates had significant requirements in the heart, we utilized the established *Drosophila* heart model and cardiac-specific RNAi KD. First, the nine candidate genes were assigned their respective *Drosophila* homologs; seven out of nine of the human HLHS candidate genes had *Drosophila* orthologs (*Figure 1B*; *Hu et al., 2011*). The *Drosophila* Gal4-UAS system (*Brand and Perrimon, 1993*) was used to test candidate genes for their role in heart function using temporal and/or spatial KD via RNAi. The *Hand[4.2]*-Gal4 driver was used for initial screening because it is a strong post-mitotic and heart-specific driver, which is expressed throughout life in the cardiomyocytes (CMs) and pericardial cells (PCs) (*Han and Olson, 2005*; *Han et al., 2006*; *Figure 1C*). Three-week-old (mid-adult stage) female flies were used to test the seven candidate genes. *Hand[4.2]*-Gal4 KD of *capt* (actin binding protein, negatively regulating actin filament assembly), *Dgkepsilon* (Diacyl glycerol kinase, DGKE), and *Chchd3/6* (Mitochondrial inner membrane protein of the MICOS complex, required for fusion) produced defects in the fly hearts, such as reduced cardiac output, reduced fractional shortening, and arrhythmicity (*Figure 1B*). Of those, *Chchd3/6* KD gave the most severe cardiac defects with strongly reduced fractional shortening, a measure of cardiac contractility. Systolic rather than diastolic diameter was increased, which suggests systolic dysfunction (*Figure 1—figure supplement 1A–C*). Since reduced contractility was previously shown in animals with reduced mitochondrial gene expression, we hypothesized *Chchd3/6* KD may reduce contractility via a role in mitochondrial function (*Bhandari et al., 2015*; *Martínez-Morentin et al., 2015*; *Tocchi, 2015*).

To test how early the cardiac phenotype of *Chchd3/6* KD manifests in adult stages, 1-week-old *Hand[4.2]*-Gal4 KD of *Chchd3/6* flies were examined, using several independent RNAi lines for *Chchd3/6*. These flies also had reduced fractional shortening, that is, reduced contractility due to systolic dysfunction (*Figure 1D–F*). This phenotype was observed in cardiac assays of intact flies (see Materials and methods; *Figure 1—figure supplement 1D–F*), as well as in the semi-intact adult heart preparation that lacks neuronal inputs (SOHA; *Fink et al., 2009*). To further validate a cardiac-specific role for *Chchd3/6*, as opposed to non-autonomous effects from other tissues, we performed KD of *Chchd3/6* using *Dot*-Gal4 (expressed in pericardial cells, PC, which also express Hand), *Mef2* (Myocyte enhancer factor 2)-Gal4 (a pan-muscle driver), or *elav*-Gal4 (a pan-neuronal driver; see Materials and methods). A large reduction in fractional shortening was only observed with the pan-muscle driver that includes cardiac muscle, but not with the PC or neuronal drivers, confirming a cardiomyocyte-autonomous effect (*Figure 1—figure supplement 1J–L*). Both C*hchd*3/6[RNAiA] and C*hchd*3/6[RNAiB] lines had the same predicted off-target gene, *Duox* but *Hand[4.2]*-*Gal4* driven KD of *Duox* had no effect on fractional shortening, confirming that the cardiac effects were due to *Chchd3/6* KD (*Figure 1—figure supplement 1M*).

## Temporal requirements for *Chchd3/6* in maintaining heart function

We next sought to understand if *Chchd3/6* has different temporal requirements for its effects on heart structure and function, since hearts of operated HLHS patients often develop reduced ejection fraction and heart failure, including the 11 H proband. First, to assess whether *Chchd3/6* could have a role in early heart development, we mined embryonic heart-specific single-cell transcriptomic data (*Vogler et al., 2021b*) and found that *Chchd3/6* was expressed in *Drosophila* cardioblasts (CBs), along with other cardiogenic factors (*tinman*, *H15*, and *Hand*) (*Figure 2A*). Next, we analyzed *Chchd3/6* mutant embryos for cardiac phenotypes. Late-stage 16–17 *Chchd[D1]* /*Chchd[DefA]* trans-heterozygous embryos were stained for Mef2 (early mesoderm/muscle-specific transcription factor) and Slit (secreted protein in the heart lumen) but did not exhibit overt cardiac specification defects (*Figure 2B*). We used the *tinD*-Gal4 driver (*tinman enhancer D Yin et al., 1997*; *Xu et al., 1998*) to test whether *Chchd3/6* KD in the dorsal mesoderm (including cardiac mesoderm) during embryonic stages 10–12 affects establishment of adult heart function. We reared *tinD*-Gal4 >C*hchd*3/6[RNAiA] flies at 29 °C throughout life to achieve high KD efficiency but did not observe reduced fractional shortening or any other functional defects, relative to controls (*Figure 2C and D*). Thus, KD of *Chchd3/6* in the embryonic cardiac mesoderm is not sufficient to impact later heart function.

To further investigate the temporal requirement of *Chchd3/6* during heart development, we made use of the temperature-dependence of Gal4-mediated KDs (less KD efficiency at 19 °C, greater KD efficiency at 29 °C; see *Figure 2E* for experimental strategy). *Hand[4.2]*-*Gal4* mediated KD of

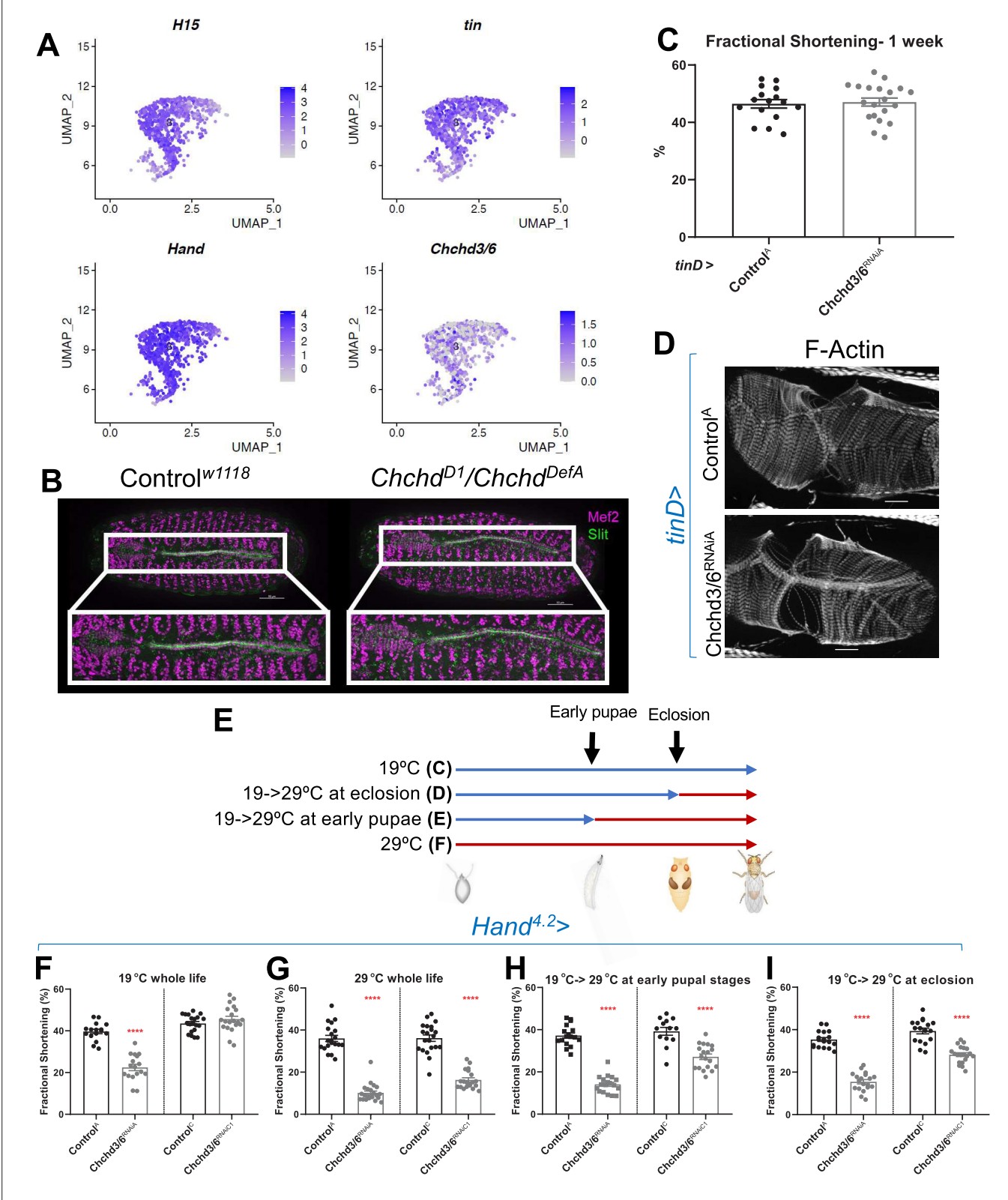

**Figure 2.** *Chchd3/6* expression is important for adult cardiac function around larval stages and early adult stages. (**A**) UMAP (uniform manifold approximation and projection) plot from CB-specific single-cell transcriptomics (*Vogler, 2021a*) showing expression of *Chchd3/6* in CBs, as identified by cardiac TFs *tin*, *H15*, and *Hand*. (**B**) Stage 16–17 embryos (late stage cardiogenesis) were collected from a *Chchd3/6* loss of function line (*Chchd^D1^*) line crossed to a *Chchd3/6* deficiency line (*Chchd^DefA^*) and stained for Mef2 (all muscle transcription factor, magenta) and Slit (secreted protein of

*Figure 2 continued on next page*

*Figure 2 continued*

the lumen, green). 50 µm scale. (**C**) *tinD* >Control[A] or>Chchd3/6[RNAiA] were reared at 29 °C and females were filmed and imaged at 1 week of age. (**A**) *tinD* >Chchd3/6[RNAiA] did not have a significant reduction in fractional shortening compared to *tinD* >Control[A] flies. (**D**) F-actin was unchanged between *tinD* >Control[A] and *tinD* >Chchd3/6[RNAiA] flies at 1 week of age; 20 µm scale. (**E**) Schematic overview of temperature shift experiments. (**F–I**) Fractional shortening measurements from 1-week-old female flies reared at (**F**) 19 °C for whole life, (**G**) at 29 °C for whole life, (**H**) 19 °C, and moved to 29 °C at early pupal stages, or (**I**) 19 °C, and moved to 29 °C once eclosed (virgin flies), Unpaired two-tailed t-test, ****p≤0.0001, error bars represent SEM.

*Chchd3/6*[RNAiA] had strong contractility defects already at 19 °C (*Figure 2F*). A weaker RNAi KD line *Chchd3/6*[RNAiC1] (see *Figure 1D–F*) caused no reduction in fractional shortening at 19 °C (*Figure 2F*), whereas at 29 °C fractional shortening was reduced similarly to the stronger KD line 19 °C. To examine different developmental windows, *Hand*[4.2]-*Gal4>Chchd3/6*[RNAiC1] flies were shifted from 19°C to 29°C at either early pupal stages or early adult stages (after eclosion) until 1 week of age when heart function was assessed (*Figure 2E*). Interestingly, both treatments caused a substantial reduction in fractional shortening, although somewhat less than at 29 °C throughout life (*Figure 2G–I*). This suggests that *Chchd3/6* is not only required during pupal development, but also at adult stages for maintaining robust heart function.

### Cardiac knockdown of *Drosophila Chchd3/6* results in severe reduction of sarcomeric actin and myosin levels

The strong heart functional defects upon *Chchd3/6* KD suggest that the contractile machinery in cardiomyocytes is severely compromised. To probe for contractile abnormalities, we examined several sarcomeric components, including filamentous (F-) Actin, Myosin heavy chain, Obscurin (present at the M-line), α-Actinin (present at the Z-line), and Sallimus (Titin component in flies, localized near the Z-line) (*Figure 3A*). The intensity of F-actin staining with phalloidin was severely diminished in the working myocardium of *Hand*[4.2]-*Gal4>Chchd3/6* KD flies (*Figure 3B–D*, arrows in B). Because *Hand*[4.2]-Gal4 expression is less in ostial cardiomyocytes (inflow valves), F-actin staining in ostial sarcomeres was minimally affected, if at all (*Figure 3B*, arrowheads). Like F-actin staining, Myosin staining was also dramatically diminished (*Figure 3E,I*) In contrast, Obscurin and α-Actinin staining was only moderately reduced (*Figure 3F, G, J and K*), and Sallimus staining was unaffected (*Figure 3H and L*). These findings suggest that loss of C*hchd*3/6 function did not abrogate the overall sarcomeric organization, but instead differentially affected the abundance of individual sarcomeric proteins. Overall, the strongly diminished F-actin and Myosin levels in cardiac myofibrils is likely responsible for the diminished contractile capacity of the ATP-dependent actomyosin network in *Chchd3/6* KD hearts.

### Actin polymerization components do not mediate sarcomeric actomyosin reduction upon cardiac *Chchd3/6* knockdown

Due to the strong reduction of F-actin levels observed with reduced *Chchd3/6* expression, we hypothesized that globular (G) to F-actin polymerization was disrupted. If *Chchd3/6* KD compromises mitochondrial ATP production in high energy-demanding CMs, the reduced ATP levels could disrupt actin polymerization and lead to reductions in F-actin and other sarcomeric proteins (*Carlier et al., 1984*; *Korn et al., 1987*; *Carlier, 1998*). To test this, we reduced the cardiac expression of several actin polymerizing and depolymerizing genes (*Figure 3—figure supplement 1A*). Cardiac KD of *Arp2/3*, *gel*, *Chd64*, *WASp*, and *TM1* caused slightly reduced fractional shortening, but not as severe as with *Chchd3/6* KD (*Figure 3—figure supplement 1A, B*). Moreover, cardiac KD of *Arp*, *Chd64*, or *WASp* resulted in substantial myofibrillar disorganization, including gaps, but did not appear to produce the *Chchd3/6* KD-like reduction in sarcomeric F-actin levels (WASp example shown in *Figure 3—figure supplement 1C*). Overall, KD of any of these genes involved in F-actin polymerization could not recapitulate the reduced myocardial F-actin intensity with normal sarcomeric patterning seen with *Chchd3/6* KD. Therefore, it appears unlikely that defects in actin polymerization mediate the effects of *Chchd3/6* KD.

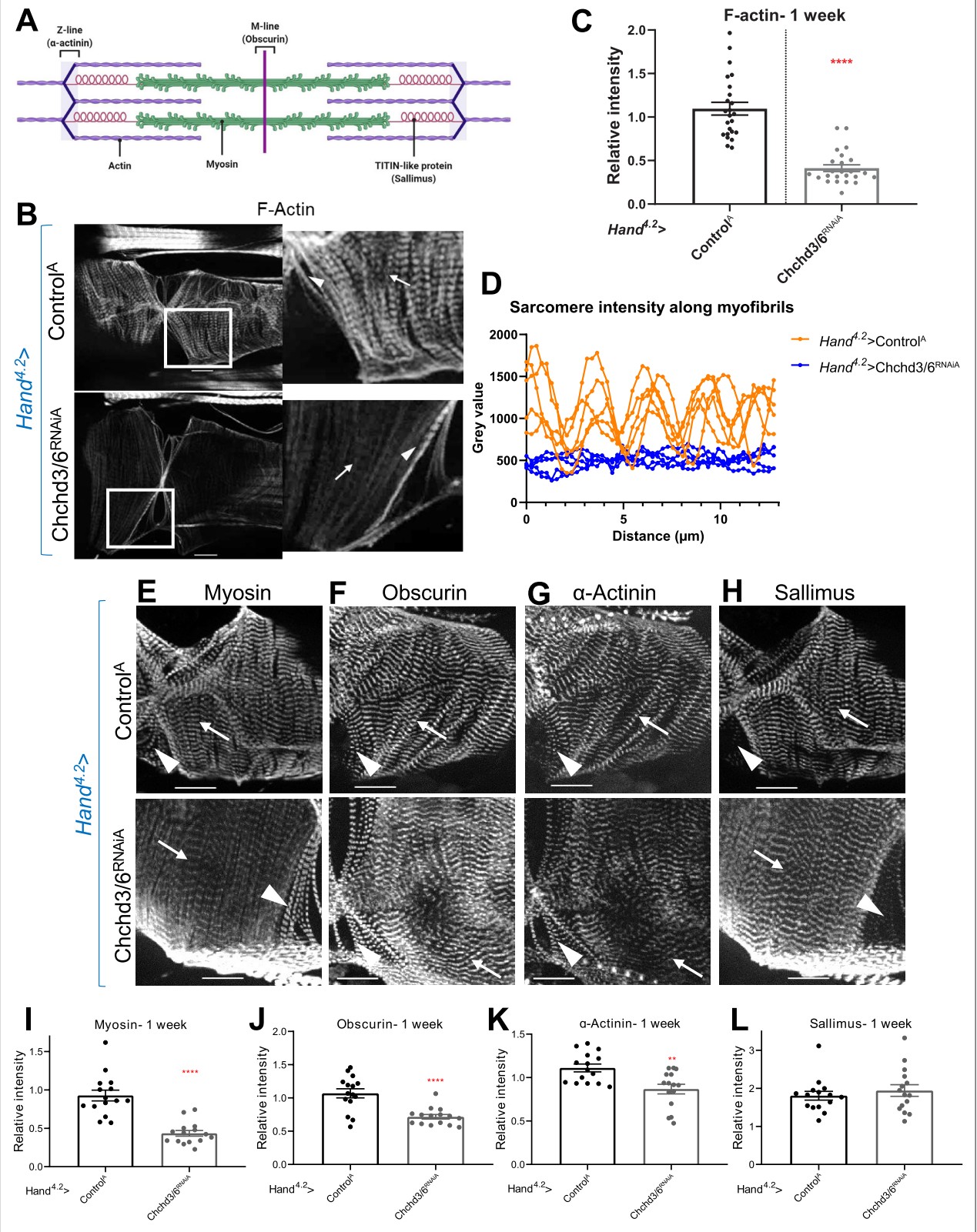

**Figure 3.** Cardiac tissue from heart-specific *Chchd3/6* KD flies exhibit reduced and altered sarcomeric proteins in the myocardial tissue. (**A**) Schematic of sarcomeric protein distribution inside myofibrils (image created with BioRender.com). (**B**) F-actin staining in 1-week-old female *Drosophila* hearts with *Hand*[4.2]-*Gal4* KD of *Chchd3/6*. Arrowheads indicate ostial myofibrils and arrows point to myocardial myofibrils (non-ostial). (**C**) F-actin intensity measured as mean gray value (gray value/# of pixels) along myocardial myofibrils relative to mean gray value of ostial myofibrils. (**D**) Mean intensity of F-actin along

*Figure 3 continued on next page*

Figure 3 continued

individual myofibrils. One-week-old *Drosophila* hearts with *Hand*<sup>4.2</sup>-*Gal4* driven KD of control or *Chchd3/6* stained for antibodies against (**E**) Myosin, (**F**) Obscurin, (**G**) α-Actinin, or (**H**) Sallimus. Arrowheads indicate ostial myofibrils and arrows point to working cardiomyocyte tissue (non-ostial). (**I–L**) Mean fluorescence intensity along myocardial myofibrils relative to ostia myofibrils in 1-week-old *Hand*<sup>4.2</sup>-*Gal4*>CHCHD3/6<sup>RNAiA</sup> adults stained for sarcomeric proteins (**I**) Myosin, (**J**) Obscurin, (**K**) α-Actinin, or (**L**) Sallimus. Unpaired two-tailed t-test, **p≤0.01, ****p≤0.0001; error bars represent SEM. 20 μm scale.

The online version of this article includes the following source data and figure supplement(s) for figure 3:

**Figure supplement 1.** KD of actin polymerizing and depolymerizing genes did not recapitulate the phenotype of Chchd3/6 KD.

**Figure supplement 1—source data 1.** Candidate genes involved in polymerization/de-polymerization of F-actin phenotypes upon KD using a *Hand*<sup>4.2</sup>-Gal4;tdtK<sup>attP2</sup> driver, measured at 1 week of age.

## *Chchd3/6* knockdown in flight or heart muscles results in defective mitochondria

Next, we hypothesized that the reduced contractile capacity and altered F-actin and Myosin in *Chchd3/6* KD hearts was due to reduced mitochondrial function. We first examined mitochondrial integrity and sarcomeric actin staining in indirect flight muscles (IFMs), since their mitochondria are easily visualized due to their large size. Upon *Chchd3/6* KD in IFMs using the pan-muscle driver *Mef2*-Gal4, we observed reduced F-actin staining and diminished sarcomere pattern definition (*Figure 4A*), similar to the cardiac phenotype (*Figure 3B*). This further indicated that the *Chchd3/6* KD phenotype is not specific to cardiac tissue, but likely affects all muscles. We then examined mitochondrial integrity upon *Chchd3/6* KD in IFMs expressing Mito::GFP (complex IV), and with antibodies against ATP synthase (complex V). Strikingly, Mito::GFP and ATP synthase staining revealed mitochondrial fission-fusion defects (*Figure 4B–E*), which is suggestive of an imbalance between fusion and fission.

Next, we assayed ATP synthase staining in *Hand*<sup>4.2</sup>-*Gal4*>*Chchd3/6*<sup>RNAiA</sup> hearts. We again observed mitochondrial fission-fusion defects, along with reduced F-actin and ATP synthase staining, relative to controls (*Figure 4F–I*). Furthermore, *Hand*<sup>4.2</sup>-*Gal4*, Mito::GFP >*Chchd3/6*<sup>RNAiA</sup> hearts also exhibited mitochondrial fission-fusion defects and reduced intensity of Mito::GFP (*Figure 4G–I*). Taken together, these data show that *Chchd3/6* KD disrupts cardiac mitochondrial morphology.

## Mitochondrial fission-fusion genes, *Drp1* and *Opa1*, interacted genetically with *Chchd3/6*

To explore the role of mitochondrial fission-fusion defects in cardiac *Chchd3/6* KD, we conducted genetic interaction experiments. In *Drosophila*, *dynamin related protein 1* (*Drp1*) promotes mitochondrial fission, and *optic atrophy 1* (*Opa1*) promotes the fusion of the inner mitochondrial membrane (*Dorn and Kitsis, 2015*). We used KD and overexpression (OE) of these genes in conjunction with a *Chchd3/6* sensitizer line we generated, *Hand*<sup>4.2</sup>-*Gal4,tdtK;Chchd3/6*<sup>RNAiC1</sup>, which at 25 °C, but not at 21 °C, exhibits significant contractility deficits, as measured by fractional shortening (*Figure 1*). At 21 °C, cardiac KD of *Drp1* or *Chchd3/6* alone had little effect, except for a slight dilation with *Chchd3/6* KD (*Figure 4J*, *Figure 4—figure supplement 1A and B*); however, the combined KD reduced contractility considerably, indicative of a genetic interaction (*Figure 4J*, *Figure 4—figure supplement 1C*). In addition, F-actin staining in the double KD was diminished and even disorganized, compared to either KD alone (*Figure 4—figure supplement 2A*). In contrast, *Drp1* OE combined with *Chchd3/6* KD at 25 °C did not alter the moderate decrease in contractility or F-actin staining (*Figure 4K*; *Figure 4—figure supplement 1F*, *Figure 4—figure supplement 2B*), but did revert the dilation due to *Chchd3/6* KD (*Figure 4—figure supplement 1D and E*).

In contrast to *Drp1*, *Opa1* KD already at 21 °C strongly exhibited systolic dysfunction, which surprisingly was slightly reversed when combined with *Chchd3/6* KD, thus indicative of a genetic interaction (*Figure 4L*, *Figure 4—figure supplement 1G–I*), possibly due to a dual role of *Opa1* (*Pernas and Scorrano, 2016*). F-actin staining was also diminished by *Opa1* KD (*Figure 4—figure supplement 2C*). OE of *Opa1* at 25 °C on its own did not affect contractility (*Figure 4M*), although the hearts were somewhat constricted (*Figure 4—figure supplement 1J and K*); however, combining *Opa1* OE and *Chchd3/6* KD contractility and F-actin staining were further reduced than *Chchd3/6* KD alone (*Figure 4M*, *Figure 4—figure supplement 2D*), although this interaction did not reach significance (*Figure 4—figure supplement 1L*). In summary, based on heart contractility and F-actin staining results, we found significant genetic interactions between mitochondrial fission-fusion genes

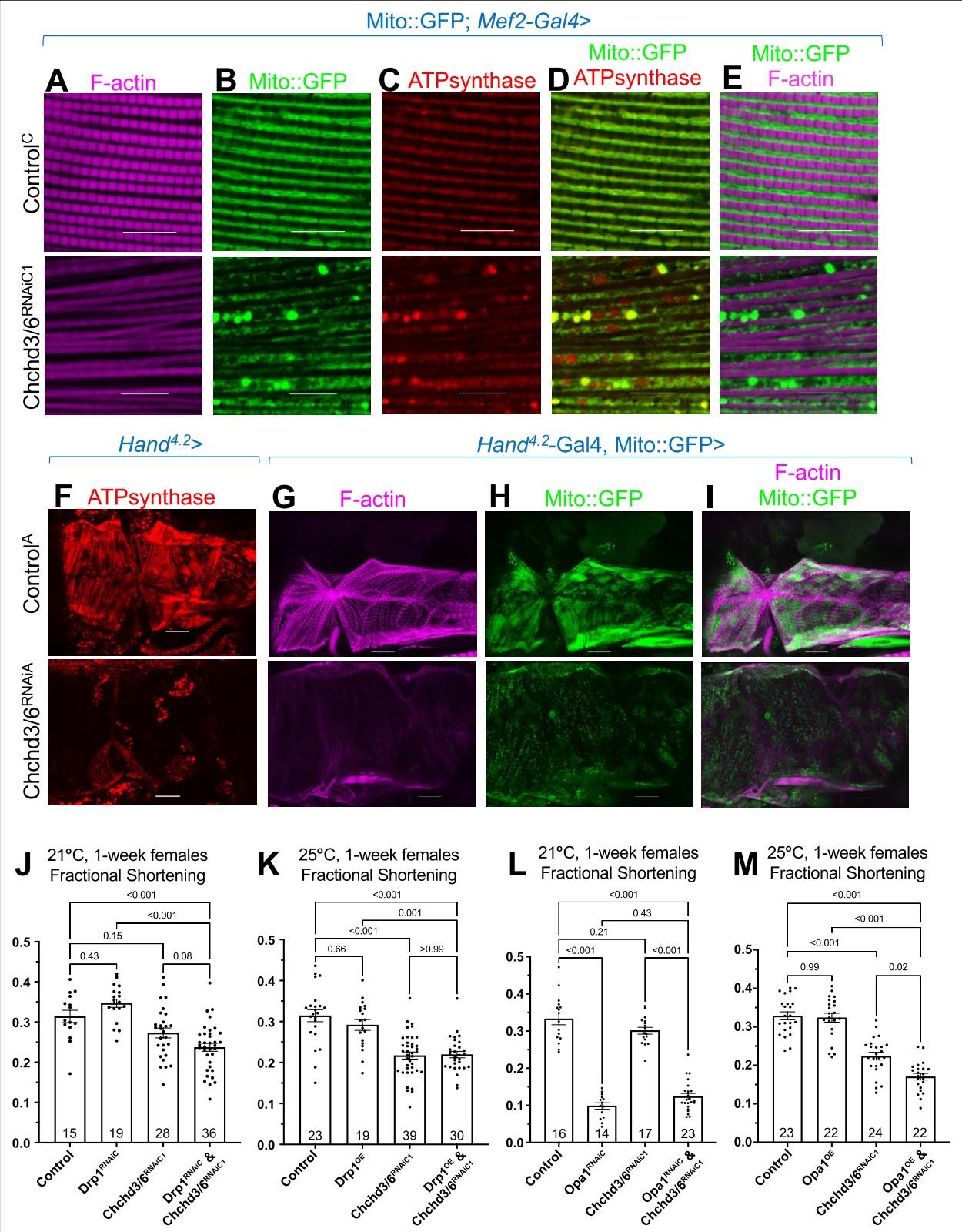

**Figure 4.** Mitochondrial fission-fusion defects were observed in cardiac *Chchd3/6* KD. (**A–E**) Visualization of F-actin and mitochondria in *Drosophila* indirect flight muscles (IFMs). 1–2 day-old male *Drosophila* IFMs with Mito::GFP; *Mef2*-Gal4 stained for (**A**) F-actin, (**B**) GFP (Mito::GFP (GFP tagged COX8A)), (**C**) ATP synthase, (**D**) merged image of B+C, and (**E**) merged image of A+B,10μm scale. (**F**) *Hand*[4.2]>Chchd3/6[RNAiA] heart tissue at 1 week of age. (**G–I**) F-actin and Mito::GFP staining in 1-week-old female hearts using the *Hand*[4.2]-*Gal4*; Mito::GFP driver, 20 μm scale. (**J–M**) Fractional shortening

*Figure 4 continued on next page*

*Figure 4 continued*

results are displayed from 1-week-old females with manipulation of mitochondrial fission-fusion genes and *Chchd3/6^RNAiC1^*. (**J**) *Chchd3/6* and *Drp1* KD at 21 °C had little effect on their own, but in combination caused a reduction in fractional shortening, displaying a significant genetic interaction (*Figure 3—figure supplement 1C*). (**K**) At 25 °C *Chchd3/6* KD reduced fractional shortening substantially, whereas *Drp1* OE by itself or in combination with *Chchd3/6^RNAiC1^* had no effect, thus no genetic interaction was observed (*Figure 3—figure supplement 1F*). (**L**) Even at 21 °C, *Opa1* KD drastically reduced contractility, which in combination with *Chchd3/6* KD slightly improved, which resulted in a significant genetic interaction (*Figure 3—figure supplement 1I*). (**M**) At 25 °C, *Opa1* OE (**M**) had no effect, but in combination with *Chchd3/6* KD contractility was further reduced significantly, although the interaction p-value did not reach significance (*Figure 3—figure supplement 1L*). One-way ANOVA with multiple comparisons shows mean with SEM and associated p-values. Sample size is shown at bottom of each bar.

The online version of this article includes the following figure supplement(s) for figure 4:

**Figure supplement 1.** Genetic interactions between mitochondrial fission-fusion genes and *Chchd3/6* KD.

**Figure supplement 2.** F-actin staining of the fly hearts in the interaction experiments involving mitochondrial fission-fusion genes and *Chchd3/6* KD.

and *Chchd3/6*, thus supporting the idea that compromised MICOS complex genes affect mitochondrial fission/fusion mechanisms.

## Mitochondrial ATP synthase (complex V) KD causes contractile dysfunction and diminished sarcomeric F-Actin staining similar to *Chchd3/6* KD

To determine whether KD of other mitochondrial genes impaired contractility and sarcomeric F-actin accumulation similar to that of *Chchd3/6* KD, we screened RNAi lines from different mitochondrial functional groups (FlyBase.org GO term mitochondrion: 0005739) using the *Hand^4.2^-Gal-4*; tdtK driver for high-throughput heart imaging analysis (see Materials and methods; *Vogler, 2021a*). Cardiac KD of 17 of the 21 mitochondrial genes tested displayed reduced fractional shortening, most commonly due to systolic dysfunction. However, only KD of *Opa1* and ATP synthase subunits reduced both fractional shortening and F-actin staining (*Figure 5*, *Figure 5—figure supplement 1A*). Remarkably, F-actin staining in *Hand^4.2^-Gal4*, tdtK >ATPsynβ/B^RNAi^ hearts resembled that of *Chchd3/6* KD, that is weakly stained myocardial myofibrils relative to ostial myofibrils (*Figure 5E*). These findings suggested that the heart dysfunction observed upon *Chchd3/6* KD may be mediated via defects in ATP synthase.

## ATP production is reduced upon *Chchd3/6* knockdown

Disrupted mitochondrial organization and reduced staining of OXPHOS components likely impacts ATP production and could explain cardiac functional and structural defects. We therefore directly measured ATP concentration in 1-week-old female hearts with cardiac *Chchd3/6* KD. We observed reduced ATP levels upon *Chchd3/6* KD compared to controls, and these ATP levels were similar to those measured in response to cardiac *ATP-synβ* KD (*Figure 5D*). This further strengthens the hypothesis that the cardiac functional deficits in contractility induced by *Chchd3/6* KD are due to mitochondrial defects that considerably reduce ATP levels in the heart necessary to build and maintain myofibrils.

## *Chchd3/6* knockdown in all muscle cells is lethal or reduces climbing ability

To further characterize the impact of *Chchd3/6* KD induced mitochondrial defects on muscle function, we assessed locomotive ability. When reared throughout development at 25 °C, Mito::GFP; *Mef2* >C*hchd*3/6^RNAiA^ flies were pupal lethal. However, with a moderate strength RNAi line (Mito::GFP; *Mef2* >CHCHD3/6^RNAiC1^) flies did eclose, but with reduced viability, especially for males (*Figure 5— figure supplement 1B*). Flies are negatively geotactic and will rapidly climb up the sides of a vial when tapped down. In 1 week of age, male and female Mito::GFP; *Mef2* >*Chchd3/6*^RNAiC1^ flies, this activity was greatly reduced compared to controls (*Figure 5—figure supplement 1C*), supporting the hypothesis that *Chchd3/6* KD reduces muscle function.

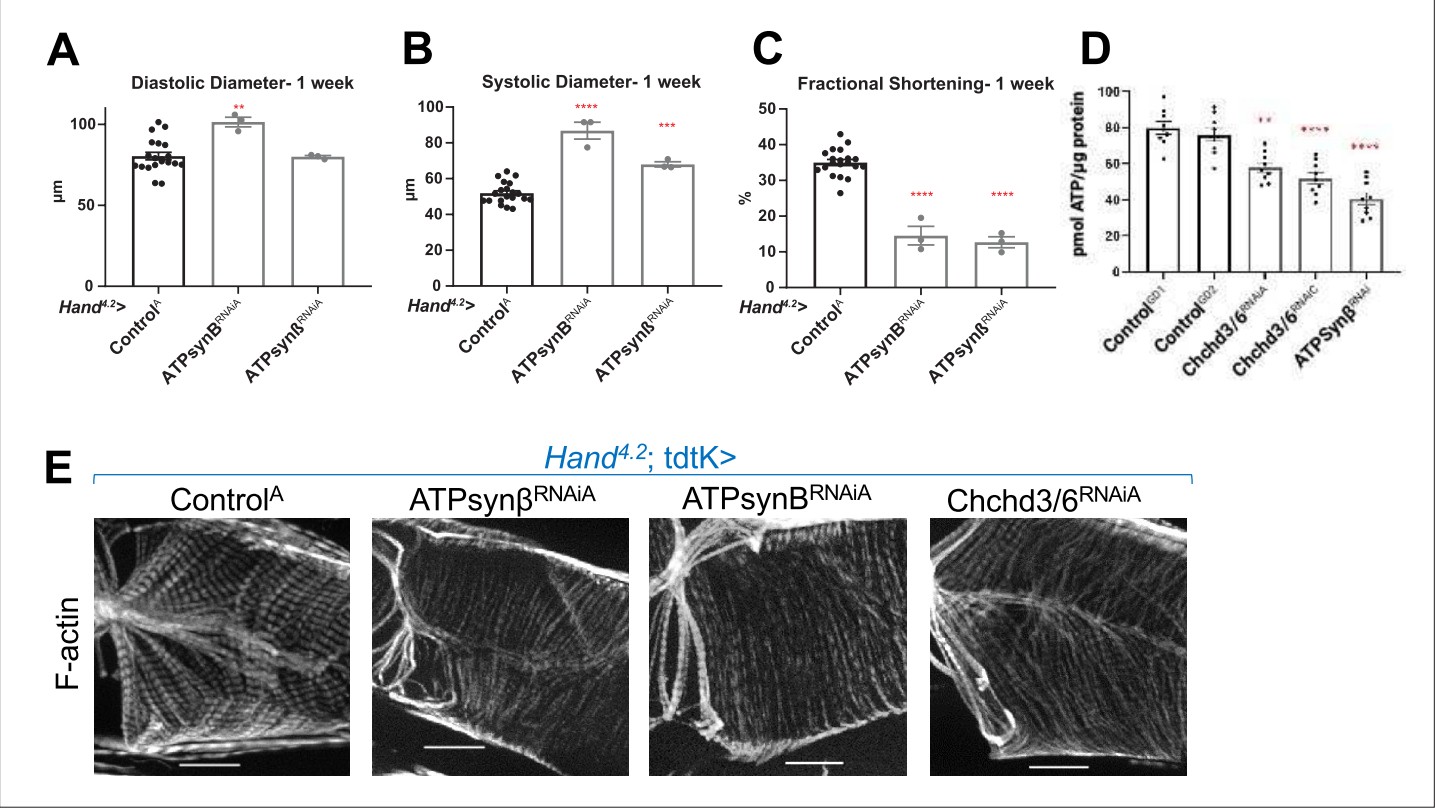

**Figure 5.** KD of ATP synthase subunit B and beta reduced both fractional shortening and F-actin staining. (**A–C**)*Hand*[4.2]*-Gal4*; tdtK driven KD of ATP synthase subunits at 1 week of age measuring (**A**) diastolic diameter, (**B**) systolic diameter, and (**C**) fractional shortening. Data is plotted as ±SEM and significance indicated relative to Control[GD2]. ****p ≤0.0001, **p ≤0.01. (**D**) Quantification of ATP levels from hearts of 1-week-old flies (10–12 hearts per sample). ATP measurements were plotted relative to protein content. (**E**) One-week-old *Hand*[4.2]*-Gal4*; tdtK driven KD of ATP synthase subunits with altered F-actin (*Chchd3/6* KD is depicted to contrast the structural phenotypes). Statistical differences were calculated by one-way ANOVA followed by Tukey's *post hoc* test for multiple comparisons.

The online version of this article includes the following figure supplement(s) for figure 5:

**Figure supplement 1.** Cardiac KD of ATP synthase components disrupted or diminished F-actin staining.

## Knockdown of *SAMM50* ortholog and *Mitofilin* causes cardiac defects, and SAMM50 genetically interacts with CHCHD3/6

To further explore the role of the MICOS complex to maintain myofibrillar structure, we tested five MICOS complex-associated components for their requirement in cardiac contractility and sarcomeric F-actin levels (*Figure 6A*). *Hand*[4.2]*-Gal4*-mediated KD of *IMMT* (*Mitofilin*) or *SAMM50* (*CG7639*) resulted in a significant reduction in fractional shortening, due to systolic dysfunction, mimicking *Chchd3/6* KD defect in contractility, whereas orthologs of *APOOL* (*Mic26/27*), *MICOS13* (*QIL1*), or *MICOS10* (*CG12479*) KD did not exhibit significant effects (*Figure 6B*; *Figure 6—figure supplement 1A,B*). Of note, none of these five MICOS-associated components displayed detectable reduction in sarcomeric F-actin staining upon KD (*Figure 6—figure supplement 1C*).

Subsequently, we tested for genetic interactions between *Chchd3/6* and MICOS-associated components, since protein-protein interactions among them have been previously described (*Ding et al., 2015*; *Li et al., 2016*; *Tang et al., 2020*). To test for interactions, we generated a *Hand*[4.2]*-Gal4*, tdtK; *Chchd*[D1/+] heterozygote mutant sensitizer line that had no noticeable cardiac abnormalities (*Deng et al., 2016*; *Figure 6C*). *Hand*[4.2]*-Gal4*, tdtK; *Chchd*[D1/+] crossed to Mitofilin or *CG7639* (*SAMM50*) RNAi did not further reduce fractional shortening beyond what was observed in response to KD of the individual genes (*Figure 6—figure supplement 1D*). However, we observed an interaction in the combined *SAMM50* KD and *Chchd*[D1/+] hearts, where F-actin levels were also strikingly reduced compared to the single KD, similar the *Chchd3/6* KD (*Figure 6D*). This suggests that there is

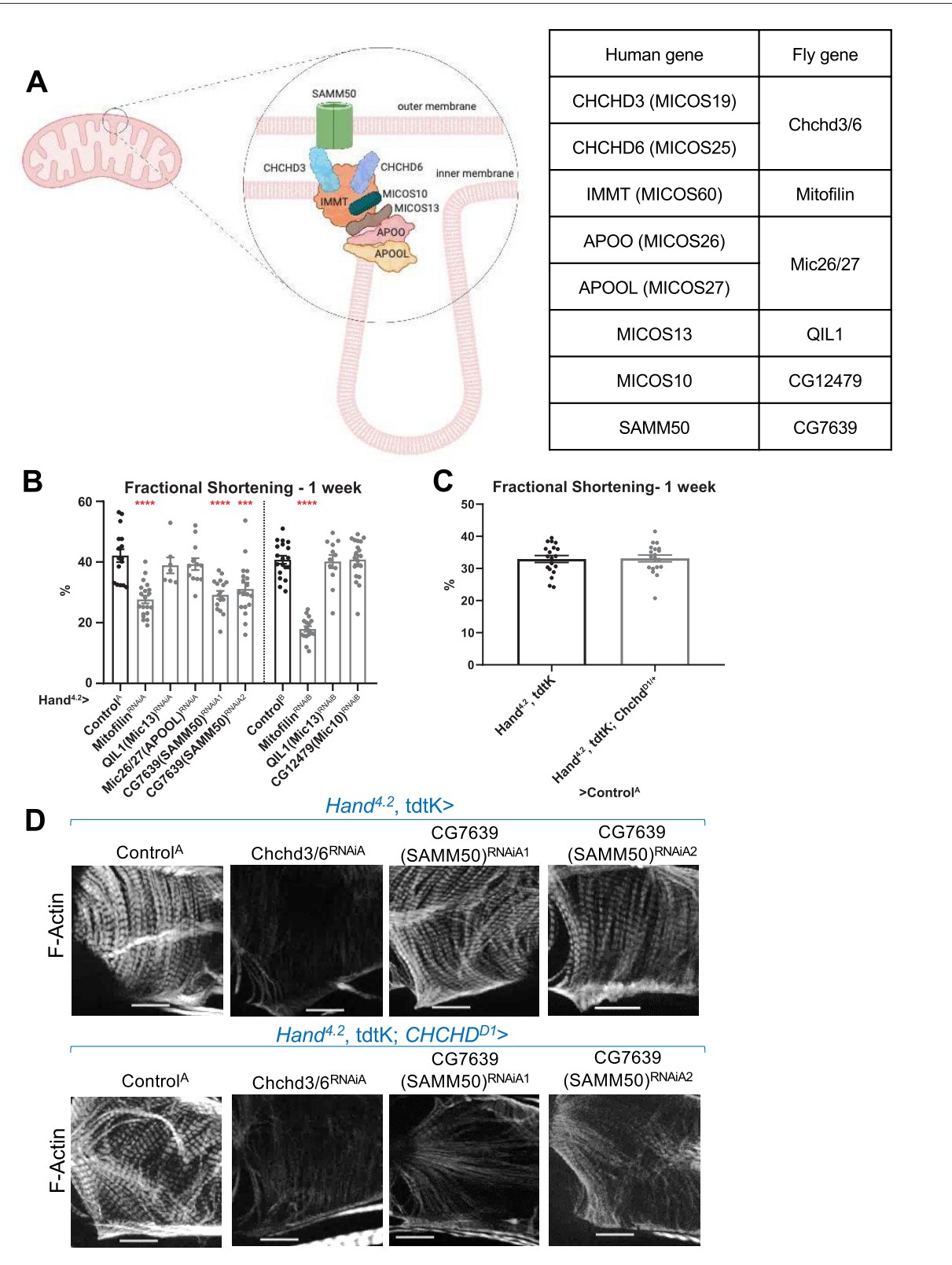

**Figure 6.** Assessment of other MICOS subunits in the *Drosophila* heart. (**A**) Schematic of the MICOS complex and SAM50. Human MICOS subunits and their respective *Drosophila* homologs are listed (image created with BioRender.com). (**B**) Fractional shortening measured from 1-week-old female flies with KD of individual MICOS subunits and *Sam50* using a *Hand^{4.2}-Gal4* driver. Unpaired two-tailed t-test, ***p≤0.001, ****p≤0.0001; error bars represent

*Figure 6 continued on next page*

**Figure 6 continued**

SEM. (**C**) *Hand^{4.2}-Gal4*, tdtK; *Chchd^{D1/+}* line was crossed out with Control^A. Unpaired two-tailed t-test, error bars represent SEM. (**D**) 1 week old F-actin-stained *Drosophila* hearts with or without heterozygous loss-of-function *Chchd^{D1/+}* in the background, 20 µm scale.

The online version of this article includes the following source data and figure supplement(s) for figure 6:

**Source data 1.** Human MICOS subunits and their respective *Drosophila* homologs.

**Figure supplement 1.** Heart function and F-actin staining in flies with KD of other components of MICOS and SAMM50.

**Figure supplement 2.** Cardiac cell proliferation and contractility assay following MICOS complex subunit KD.

a threshold requirement of MICOS/SAMM50, which when reached induces reduced contractility AND diminished F-Actin levels.

## Knockdown of MICOS subunits impairs proliferation and oxygen consumption of human iPSC-derived cardiomyocytes

We next tested the effects of KD of *CHCHD3/6* and other MICOS-associated components in human cardiomyocytes (CMs) derived from iPSC (hiPSC-CM; *Cunningham et al., 2017*, *Yu et al., 2018*). Since reduced CM proliferation is hypothesized to be a major contributing factor for the etiology of HLHS (*Gaber et al., 2013*; *Liu et al., 2017*; *Theis et al., 2020*), we focused on proliferation as our readout. We used small interfering RNA (siRNA) (as in *Theis et al., 2020*) to KD genes in hiPSC-CMs. We found that KD of *CHCHD6* and *CHCHD3*, as well as all other MICOS subunits and *SAMM50*, significantly reduced their proliferation in an EdU incorporation assay (*Figure 6—figure supplement 2A,B*), supporting a potential link for *CHCHD6* and other MICOS subunits in HLHS pathogenesis.

Moreover, oxygen consumption rate (OCR), was significantly decreased in the combined KD of *CHCHD3* and *CHCHD6* in hiPSC-CMs 60 min after inhibition of ATP synthase by oligomycin treatment (*Figure 6—figure supplement 2C*). Similar to the fly heart results, staining of sarcomeric F-actin along the fiber (yellow line in Supp. ) was reduced by co-KD of *CHCHD3* and *CHCHD6* (*Figure 6—figure supplement 2E*).

## Testing of candidate genes prioritized in HLHS probands with *CHCHD3* or *CHCHD6* variants reveals novel genetic interactors

Since we found an essential role for *Chchd3/6* in establishing heart structure and function in the *Drosophila* heart model with possible relevance for HLHS pathology, we assessed the presence of variants in additional HLHS family trios. Among the 183 Mayo Clinic HLHS family trios and pediatric cardiac genomics consortium (PCGC) databank (*Jin et al., 2017*), there were three probands with variants in *CHCHD6* (including 11 H) and four with *CHCHD3* variants. In total, there were four noncoding variants with a RegulomeDB rank of 2 a (n=2), 2b (n=1) and 4 (n=1) providing evidence for haploinsufficiency due to predicted disruption of transcription factor binding in the presence of the variant. The remaining two missense variants, one inherited and one de novo, may alter the protein structure or function and lead to downstream functional consequences.

The relative abundance of rare, predicted damaging *CHCHD3/6* variants in the Mayo Clinic cohort, together with the postulated oligogenic nature of HLHS, led us to test for genetic interactions between CHCHD3/6 and other HLHS candidate genes. Specifically, we prioritized candidate genes with rare coding and regulatory variants identified in HLHS probands who also carried *CHCHD3*- or *CHCHD6*-variants. With the *Chchd3/6* sensitizer line, *Hand^{4.2}-Gal4, tdtK; Chchd3/6^{RNAiC1}*, which at 21 °C does not exhibit significant contractility deficits (*Figure 7A*), we screened 120 RNAi lines representing 60 candidate HLHS genes for genetic interactions and identified three hits that had contractility defects only when co-knocked down with *Chchd3/6^{RNAiC1}* at 21 °C (*Figure 7B*). *Cdk12* (human ortholog: *CDK12*) has been shown to activate RNA polymerase II to regulate transcription elongation (*Bartkowiak et al., 2010*), *goliath* (human ortholog: *RNF149*) encodes an E3 ubiquitin ligase that localizes to endosomes (*Yamazaki et al., 2013*) and *β-Spectrin* (human ortholog: *SPTBN1*) is a scaffolding protein that links the actin cytoskeleton to the plasma membrane. KD of *Cdk12* in combination with *Chchd3/6* KD also led to greater lethality of elcosed flies at 1 week-of-age compared to individual gene KD (*Figure 7C*). While interestingly, we found that co-KD of *β-Spectrin* and Chchd3/6^{RNAiC1} at 21 °C was the only combination that also diminished F-actin and Myosin staining, similar to co-Chchd3/6^{RNAiA} KD (*Figure 7D*). In summary, our approach using a sensitized screening strategy to interrogate genetic

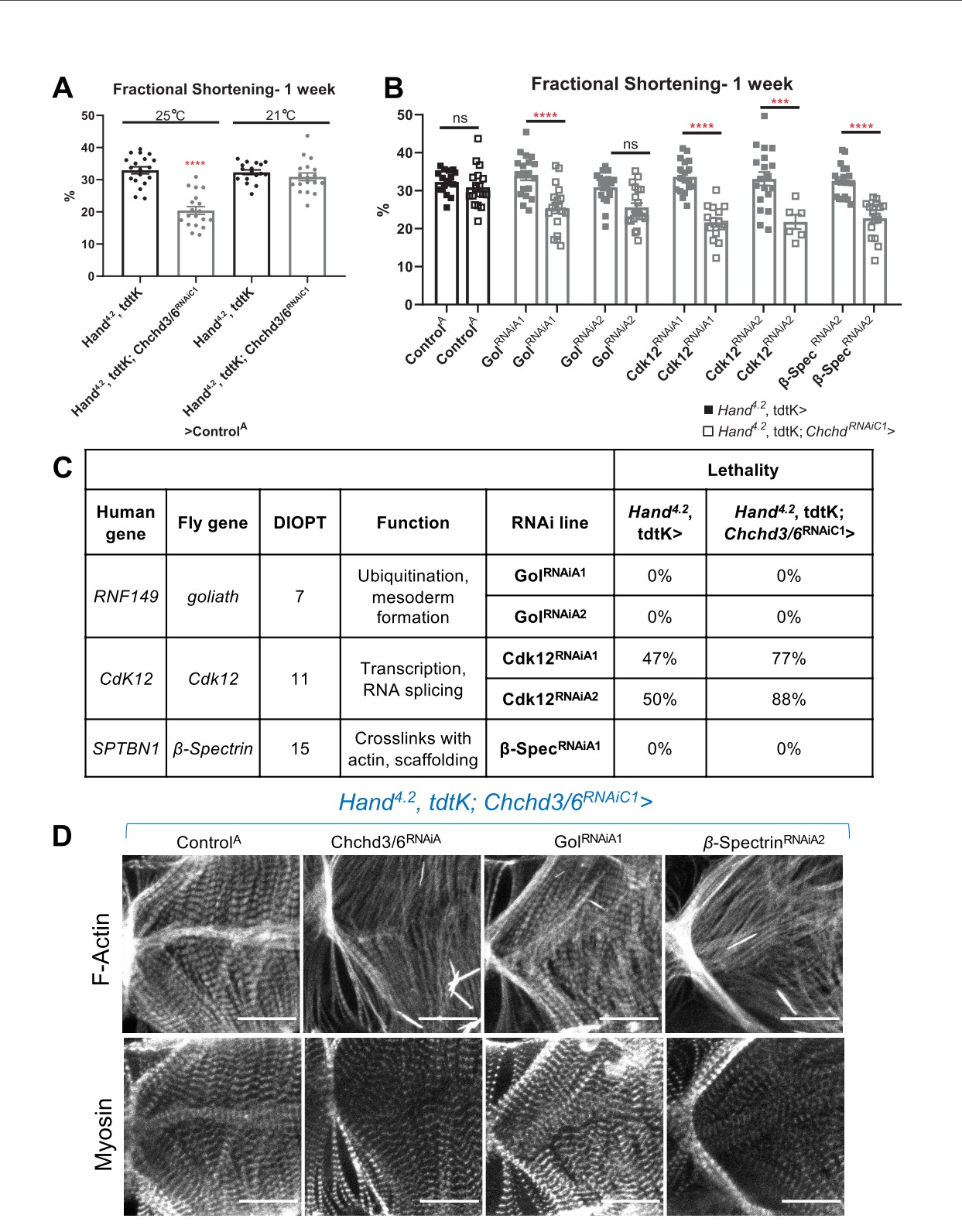

**Figure 7.** HLHS CHCHD3 and CHCHD6 family-based gene interaction screen reveals three hits. (**A**) A Hand4.2-Gal4, tdtK; Chchd3/6RNAiC1 sensitizer line show reduced fractional shortening at 25°C, which is no longer significant at 21°C. Unpaired two-tailed t-test, ****p≤ 0.0001; error bars represent SEM. (**B**) Genetic interaction of Chchd3/6 and prioritized HLHS candidates. Two-way ANOVA with Tukey's multiple comparisons test, only statistical

*Figure 7 continued on next page*

*Figure 7 continued*

comparisons between the same RNAi lines are shown; ***p≤ 0.001, ****p≤ 0.0001; error bars represent SEM. (**C**) Functional overview of human and *Drosophila* orthologs. KD of Ckd12 with Chchd3/6 KD led to increased lethality of eclosed flies by 1 week-of-age.

The online version of this article includes the following source data for figure 7:

**Source data 1.** Functional overview of human and *Drosophila* orthologs.

interactions between patient-specific candidate genes is a powerful tool to identify novel interactions that are potentially involved in the oligogenic ethology of HLHS and other CHDs.

## Discussion

HLHS is characterized by a small left heart, including reduced left ventricle size and mitral and/or atrial atresia or stenosis, and aortic hypoplasia, collectively obstructing systemic blood flow (*Tchervenkov et al., 2006*). As a consequence, newborns cannot sustain systemic blood flow for more than a few days and therefore require treatment soon after birth. There is a need for improved therapies to treat HLHS patients, and this requires a better understanding of the biology behind HLHS pathogenesis. Here, we probed the genetic basis of HLHS using WGS and powerful bioinformatic gene variant prioritization in a large cohort of HLHS proband-parent trios combined with model system validation.

The 11 H family was prioritized because of consanguinity, implicating a homozygous recessive mode of inheritance that resulted in a short list of nine candidate genes. These candidate genes were probed in *Drosophila* and iPSC-CMs for a potential role in cardiomyocyte development and function, to gain new insights into HLHS and CHDs in general. Among these HLHS gene candidates, we focused on *CHCHD3/6*, which has not been previously studied in the heart, and which had striking cardiac functional and structural defects in *Drosophila*. Specifically, the preliminary gene screen demonstrated that *Chchd3/6* cardiac-specific KD caused reduced contractility and decreased sarcomeric F-Actin and Myosin staining.

Our data suggest that *Chchd3/6* is necessary during larval and early adult stages to maintain contractility in the adult heart. This is relevant since patients with HLHS have both structural heart disease and risk for later myocardial failure (*Theis et al., 2015b*). The prevailing 'no flow, no grow' hypothesis for HLHS pathogenesis surmises that reduced blood flow in the fetal heart causes underdevelopment of the left ventricle (*Goldberg and Rychik, 2016*; *Grossfeld et al., 2019*). A reduced ability for the heart to contract in utero, due to reduced *CHCHD6* activity, could contribute to decreased ventricular blood flow in the embryo, resulting in an abnormally small left ventricle. Moreover, reduced *CHCHD6* activity could compromise right ventricular function later in life (*Theis et al., 2015a*; *Theis et al., 2015b*). In fact, the 11 H proband exhibited mildly reduced right ventricular ejection fraction several years after successful surgical palliation. Consistent with our model system, *CHCHD6* deficiency could result in cumulative impairment of mitochondrial function, leading to contractile dysfunction (*Sun et al., 2016*). Why a mitochondrial defect would have a preferential effect on the left ventricle is still an enigma. We speculate that some of the patient-specific variants that potentially contribute to this likely polygenic disease are in genes that may have a higher expression level or functional importance in the left ventricle, thus in combination with MICOS variants preferentially affecting left-ventricular growth and differentiation, leading to decreased contractility, then again compounded by impaired blood flow feeding back to diminishing growth. Future studies investigating the polygenic basis of HLHS are needed to address this question.

*Chchd3/6* KD in the fly heart led to mitochondrial fission-fusion defects, with reduced ATP synthase (complex V) levels, and consequently impaired ATP production. It has previously been reported that *CHCHD3* KD in HeLa cells resulted in fragmented mitochondria that was due to improper mitochondrial fusion (*Darshi et al., 2011*). It has also been demonstrated in yeast that individual or combinatorial loss of MICOS complex proteins disrupt cristae morphology (*Friedman et al., 2015*), thus suggesting a mechanism by which *CHCHD3/6* loss could mediate HLHS pathogenesis. Furthermore, we identified a genetic interaction between SAMM50 and CHCHD3/6 that leads to a contractile deficit and diminished sarcomeric F-Actin. Recent findings demonstrate that SAMM50 directly interacts mammalian CHCHD3, to mediate inner and outer membrane bridging and cristae morphology (*Tang et al., 2020*).

Our data further suggest that ETC Complex V/ ATP synthase is a potential downstream effector of *CHCHD3/6* and MICOS complex function. Individual KD of ATP synthase subunits resulted in reduced fractional shortening and reduced sarcomeric actin. As a result, we hypothesize reduced *CHCHD3/6* expression affects ETC function, specifically ATP synthase, leading to reduced ATP production. Since the MICOS complex is in involved ETC assembly in cristae, ATPase subunits may not be assembled correctly causing mitochondrial dysfunction, accompanied by reduced/abnormal mito-GFP staining (see *Figure 4B and H*). OXPHOS complex assembly has been shown to be disrupted upon MICOS depletion, and we speculate ATP synthase function may be disrupted when *CHCHD3/6* is reduced (*Cogliati et al., 2016*). Consistent with this, we observe depletion of ATP synthase levels upon *Chchd3/6* KD.

Finally, we tested a potential oligogenic basis of HLHS in our family-based *CHCHD3* and *CHCHD6* interaction screen and identified three hits that reduced fractional shortening only in conjunction with *CHCHD3/6*, but not on their own. Co-KD of *Cdk12* and *Chchd3/6* also reduced fractional shortening, and caused greater lethality relative to *Cdk12* KD alone. Cdk12 activates RNA polymerase II to regulate transcription elongation (*Bartkowiak et al., 2010*). We postulate that since *Chchd3/6* is a nuclear-encoded gene, reducing transcription with *Cdk12* KD could decrease *CHCHD3/6* levels in a background where *CHCHD3/6* activity is already compromised. Alternatively, reduced transcription of other nuclear genes associated with ATP production in combination with *Chchd3/6* KD could further reduce ATP levels enough to cause contractility defects. In support of this, a study examining the effects of RMP (RNA polymerase II subunit 5-mediating protein) found that mice with cardiac-specific *Rpm* KO exhibited reduced fractional shortening and ATP levels, which were attributed to a reduction in mRNA and protein levels of the mitochondrial biogenesis factor PGC1α (*Zhang et al., 2019*). The second hit, *goliath*, is an endosomal ubiquitin E3 ligase. Although goliath has been implicated in endosomal recycling (*Yamazaki et al., 2013*), its role in *Drosophila* mitophagy in vivo has not been examined. Reduced cardiac contractility with co-KD of *gol* and Chchd3/6 could result from impaired mitophagy and reduced mitochondrial biogenesis. Together, the accumulation of damaged mitochondria can reduce ATP content required for contraction (*Palikaras and Tavernarakis, 2014*; *Palikaras et al., 2015*). The third hit, *β-Spectrin*, acts as a scaffolding protein. Recent data suggests that the human ortholog, SPTBN1 (Nonerythroid spectrin *β*) influences SPTAN1 (Nonerythroid spectrin α) levels, which has a calmodulin binding domain (*Ackermann and Brieger, 2019*). Therefore, decreased *β-Spec* expression could reduce Calmodulin levels, thereby reducing contractility due to the combined reduction in $Ca^{2+}$ handling and *Chchd3/6* KD-induced reduced ATP levels.

In summary, we have identified a novel mechanism potentially involved HLHS pathogenesis, starting by analyzing WGS data from a prioritized family and large cohort of HLHS patients, followed by functional testing in vivo using the *Drosophila* heart model and in vitro using human iPSC-derived CMs. Compromised contractile capacity, diminished sarcomeric F-Actin and Myosin accumulation, and mitochondrial dysfunction in *Chchd3/6* KD *Drosophila* hearts are promising phenotypes that could contribute to early HLHS manifestations or heart failure complications later in life. Further examination of the interactions between the MICOS complex and other emerging candidate genes will identify novel gene functions and pathways that contribute to HLHS pathogenesis. Furthermore, a detailed elucidation of novel candidate genes and genetic interactions based on patient-specific rare potentially damaging variants is expected to lead to gene networks that are relevant for HLHS and other CHDs.

# Materials and methods
## Study subjects
Written informed consent was obtained for the index family and an HLHS cohort, under a research protocol approved by the Mayo Clinic Institutional Review Board ('Genetic Investigations in Hypoplastic Left Heart Syndrome', IRB #11–000114). Participants consented to providing clinical health record data, sample procurement for DNA analyses, and publication of de-identified research findings. Cardiac anatomy was assessed by echocardiography.

## Comparative genomic hybridization

To detect aneuploidy, array comparative genomic hybridization was performed using a custom 180 K oligonucleotide microarray (Agilent, Santa Clara, CA), with a genome-wide functional resolution of approximately 100 kilobases. Deletions larger than 200 kilobases and duplications larger than 500 kilobases were considered clinically relevant.

## Genomic and bioinformatics analysis of 11H family

Genomic DNA was isolated from peripheral white blood cells or saliva. WGS and variant call annotation were performed utilizing the Mayo Clinic Medical Genome Facility and Bioinformatics Core. For the family quintet, 101 base pair (bp) or 150 bp paired-end sequencing was carried out on Illumina's HiSeq 2000 or HiSeq 4000 platforms, respectively. Reads were aligned to the hg38 reference genome using BWA version 0.7.10 (http://bio-bwa.sourceforge.net/bwa.shtml) and duplicate reads were marked using Picard (http://picard.sourceforge.net). Local realignment of INDELs and base quality score recalibration were then performed using the Genome Analysis Toolkit version 3.4–46 (GATK) (*McKenna et al., 2010*). SNVs and INDELs were called across all samples simultaneously using GATK's Unified Genotype with variant quality score recalibration (VQSR) (*Poplin et al., 2018*).

Variant call format (VCF) files with SNV and INDEL calls from each family member were uploaded and analyzed using Ingenuity Variant Analysis software (QIAGEN, Redwood City, CA) where variants were functionally annotated and filtered by an iterative process. Annotated variants were subject to quality filters and required to pass Variant Quality Score Recalibration (VQSR) and have a genotype quality score ≥20. Variants were excluded if they were located in a simple repeat region identified using tandem repeats finder (*Benson, 1999*) or were found to have a minor allele frequency >1% in gnomAD v2.1 (*Karczewski et al., 2020*). Second, functional variants were selected, defined as those that impacted a protein sequence, canonical splice site, microRNA coding sequence/binding site, or transcription factor binding site within a promoter validated by ENCODE chromatin immunoprecipitation experiments (*Raney et al., 2014*). Third, using parental and sibling WGS data, rare, functional variants were then filtered for those that were homozygous recessive in the proband.

## Analysis of HLHS cohort for variants in the MICOS complex

WGS was performed on samples from 183 individuals with HLHS and 496 family members by the Mayo Clinic Medical Genome Facility or Discovery Life Sciences. SNVs and INDELs that passed quality control were subject to further filtering based upon rarity (MAF <0.01) and predicted consequence. Details about the sequencing and subsequent variant filtering to identify HLHS gene candidates have been previously described (*Theis, 2021*). Rare variants from the 183 probands were interrogated for variants in *CHCHD3* and *CHCHD6* to identify variants that arose de novo or were homozygous recessive, compound heterozygous or X-linked recessive. Next, inherited variants in these genes were analyzed, but stricter thresholds were required to identify the most damaging variants. Missense variants were required to have CADD >24 (corresponds to the upper quartile of the most damaging missense variants) and non-coding variants were required to have a Position Weight Matrix (PWM) score >0.75 from the Factorbook database (selecting for variants predicted to disrupt canonical transcription factor binding sites). In addition to the Mayo Clinic HLHS cohort, the Pediatric Cardiac Genomics Consortium (PCGC) whole exome sequencing dataset was interrogated for candidate genes in the MICOS complex in patients with CHD (*Jin et al., 2017*).

## Analysis of *CHCHD3* and *CHCHD6* variant carriers

Using robust bioinformatics algorithms as previously described (*Theis and Olson, 2022*), a broad range of both family-based Mendelian inheritance modeling and cohort-wide enrichment analyses were applied to identify additional candidate genes in HLHS probands identified to have a rare, predicted-damaging coding or regulatory variant in *CHCHD3* or *CHCHD6*.

## *Drosophila* strains and husbandry

*Drosophila* crosses were reared and aged at 25 °C, unless otherwise noted. *Drosophila* orthologs were determined using DIOPT (*Drosophila* RNAi Screening Center Integrative Ortholog Prediction Tool) which calculates the number of databases that predict orthology (out of a score of 16) (*Hu et al., 2011*). Fly stocks were obtained from Vienna *Drosophila* Resource Center (VDRC) and Bloomington

*Drosophila* Stock Center (BDSC). Lines include *Hand$^{4.2}$-Gal4* (*Han et al., 2006*), *tdtK* (*Klassen et al., 2017*), *tinCΔ4-Gal4* (*Lo and Frasch, 2001*), *Dot-Gal4* (*Kimbrell et al., 2002*), *Mef2-Gal4* (*Ranganay-akulu et al., 1998*), Mito::GFP (BDSC: 8442), *CHCHD$^{DefA}$* (BL: 26847), *CHCHD3/6$^{RNAiA}$* (VDRC: 52251), *CHCHD3/6$^{RNAiB}$* (VDRC: 105329), *CHCHD3/6$^{RNAiC1}$* (BDSC: 51157), *CHCHD3/6$^{RNAiC2}$* (BDSC: 38984), *Duox$^{RNAi}$* (BDSC: 32903), *Mitofilin$^{RNAiA}$* (VDRC: 47615), *Mitofilin$^{RNAiB}$* (VDRC: 106757), *QIL1(Mic13)$^{RNAiA}$* (VDRC: 14283), *QIL1(Mic13)$^{RNAiB}$* (VDRC: 100911), *CG12479(Mic10)$^{RNAiA}$* (VDRC: 102479), *Mic26/27(APOOL)$^{RNAiA}$* (VDRC: 31098), *CG7639(SAMM50)$^{RNAiA}$* (VDRC: 33641), *CG7639(SAMM50)$^{RNAiB}$* (VDRC: 33642), *Drp1$^{RNAiA}$* (VDRC: 44155), *Drp1$^{RNAiC}$* (BDSC: 27682), *Drp1$^{OE}$* (BDSC: 51647), *Opa1$^{RNAiB}$* (VDRC: 106290), *Opa1$^{RNAiC}$* (BDSC: 32358) and *Opa1$^{OE}$* (BDSC: 95258). *CHCHD$^{D1}$* was kindly shared by the Ge lab (*Deng et al., 2016*).

Assessment of lethality in co-KD of *CHCHD3/6* and *Cdk12* refers to percentage of surviving flies at 1 week-of-age, versus the number of flies eclosing on day 0.

## In situ heartbeat analysis

An in-situ dissection approach was used to expose the denervated beating fly heart (*Fink et al., 2009*; *Ocorr et al., 2009*; *Vogler and Ocorr, 2009*). SOHA (Semi-automated optical heartbeat analysis) was used to analyze high speed video recordings to determine heart-related parameters (*Fink et al., 2009*). Flies (n>15) were briefly anesthetized using filter paper with 10 µm FlyNap and transferred to a 10X35 mm Petri dish with Vaseline to attach the hydrophobic wing cuticle to the dish. Oxygen-ated room temperature artificial hemolymph (108 mM NaCl, 5 mM KCl, 2 mM CaCl$_2$•2H$_2$O, 8 mM MgCl$_2$•6H$_2$O, 15 mM pH 7.1 HEPES, 1 mM NaH$_2$PO$_4$•H$_2$O, 4 mM NaHCO$_3$, 10 mM sucrose, and 5 mM trehalose) was added to each dish. Flies were dissected as per *Vogler and Ocorr, 2009* and oxygenated for minimum 15 min to equilibrate. Dissected flies were filmed for 30 s using an Olympus BX63 microscope (10 X magnification), a Hamamatsu C11440 ORCA-flash4.0 OLT digital camera, and HCImageLive program. These videos were uploaded to SOHA (semi-automated optical heartbeat analysis), end diastolic and end systolic diameters were manually marked towards end of ostia, and heart-related parameters were extracted (*Fink et al., 2009*).

## In vivo heartbeat analysis

Norland #61 optical glue was placed on a 22X50 mm coverslip (one small drop for each fly). Flies (n>15) were briefly anesthetized using filter paper with 10 µm FlyNap, transferred to coverslip on indi-vidual adhesive drops with the dorsal side facing the coverglass, and cured for 30 s using ultraviolet light. The coverslip was then placed on a 10X35 mm Petri dish and secured using putty. Fly hearts were filmed for 5 s using an Olympus BX63 microscope (×20 magnification), a Hamamatsu C11440 ORCA-flash4.0LT digital camera, and HCImage Live program. All analysis was automatically processed using R (*Vogler, 2021a*).

## Adult *Drosophila* heart immunohistochemistry

Flies were dissected as per *Vogler and Ocorr, 2009* in a 10X35 mm Petri dish and EGTA was added to a final concentration of 10 mM. EGTA was removed and replaced with 4% methanol-free formalde-hyde for 20 min. Formaldehyde was removed and replaced with 1 X PBS 3 times. Fly thoraxes were removed, abdominal walls were trimmed, and excess fat around heart was removed. Hearts were then washed three times with 0.3% PBTx (Triton-X) for 15 min on a shaker. PBTx was removed and replaced with 200 µL 1° antibody solution (0.3% PBTx +1° antibody), then a small piece of Parafilm with care-fully placed over the solution to form and seal of liquid over the hearts. Dishes were incubated either (1) at 4 °C overnight or (2) at room temperature for 2 hr while shaking. Once finished incubating, Parafilm was gently removed with forceps and three 15-min washes with 0.3% PBTx were performed. PBTx was removed and replaced with 200 µL 2° antibody solution (0.3% PBTx +2° antibody), then a small piece of Parafilm was carefully placed over the solution. Dishes were incubated either (1) at 4 °C overnight or (2) at room temperature for 2 hr while shaking. Parafilm was gently removed with forceps and three 15-min washes with 0.3% PBTx were performed. PBTx was removed and replaced with 1 X PBS. Ventral cuticle with attached hearts were carefully removed individually from the Vaseline layer and transferred to a 25X75 X 1mm slide with 2 18X18 mm No. 1 coverslips glued to form a bridge and ProLong Gold antifade mounting medium (Invitrogen) in the middle. Flies were placed ventral side up

and covered with a 18X18 mm No. 1.5 coverslip, sealed with clear nail polish around the edges, and stored at room temperature for 24 hr until being moved to 4 °C.

Primary Antibodies: anti-dMef2 (1:20, gift from Dr. Bruce Paterson); anti-Slit (1:40, c555 DSHB); anti-Myosin (1:50, 3E8-3D3 DSHB); anti-Sallimus (1:100, Abcam); anti-ATP5A (1:100, Abcam 14748); anti-ATP5A1 (1:200, Invitrogen 43–9800). All Secondaries from Jackson Immuno Research Labs used at 1:500: Goat anti-Rat 594; Goat anti-Rabbit 647; Goat anti-Rabbit Cy5; Goat anti-mouse Cy3. Dyes: Phallodin 594 or 647 (1:100, Invitrogen).

### *Drosophila* indirect flight muscle dissection and immunofluorescence

Thoraxes were removed under light $CO_2$ pressure and fixed for 40 min in 5% PFA, followed by three 2-min PBS washes. IFM muscle fibers were removed using fine (#55) forceps and washed with 0.5% PBTx for 15 min, then washed with 0.1% PBTx twice for 15 min. All subsequent antibody stainings were diluted in 0.1% PBTx and incubated shaking at 4 °C overnight to penetrate the muscle tissue. IFMs were transferred to a 25X75 X 1mm slide without a bridge. ProLong Gold antifade mounting medium (Invitrogen) was added, the samples were covered with an 18X18 mm No. 1.5 coverslip, and sealed with clear nail polish around the edges.

### *Drosophila* embryo collection, fixation, and immunofluorescence

Adult flies were reared in a plastic bottle cage with a Petri dish on the bottom containing grape agar (Agar, EtOH, glacial acetic acid, grape juice) and yeast paste (yeast and $H_2O$) at 25 °C. After incubating (16 hr), embryos were carefully collected with a brush and placed in a mesh basket. Flies were washed with water, then placed in bleach for 3 min, followed by 30 s of wash with water. Embryos were removed from mesh and placed in a fixation solution (2 Heptane: 1 2 X PBS: 1 10% formaldehyde) for 25 min. Formaldehyde layer (bottom) was removed, replaced with 500 µL MetOH, vortexed, then the supernatant with vitelline membranes (middle layer) was removed, this was repeated once more. Embryos were washed with MetOH (3 rinses, followed by 1 hr on rotator). Embryos were stored in fresh MetOH at –20 °C. 1° antibody was added and tube was placed on rotator at 4 °C overnight. 1° was later removed using 3X15 min 0.4% PBTx washes. PBTx was removed, replaced with 2° antibody (diluted in PBTx), and rotated for 2 hr. 2° was later removed using 3X15 min 0.4% PBTx washes, then left in PBS at 4 °C. Since $Chchd^{D1}$ and $Chchd^{DefA}$ are both homozygous lethal at adult stages, each line was rebalanced over TM6b YFP ($Chchd^{D1}$/TM6b YFP and $Chchd^{DefA}$/TM6b YFP). The YFP lines ($Chchd^{D1}$/TM6b YFP and $Chchd^{DefA}$/TM6b YFP) were crossed out and embryos were selected against GFP to obtain only $Chchd^{D1}$/$Chchd^{DefA}$ embryos.

### Fixed sample imaging

Samples were imaged at ×10, ×25, or ×40 magnification using a Zeiss Apotome.2 Imager Z1, a Hamamatsu C11440 ORCA-Flash4.0LT digital camera, and Zeiss ZEN. In order to obtain higher resolution, confocal microscopy was performed for all immunohistochemistry experiments involving Mito::GFP and ATP synthase staining.

### Climbing assay

Flies were initially anesthetized using FlyNap, placed into five separate vials, and counted for a total at week 0 for Control C females = 178, Control C males = 125, *Chchd3/6* C females = 154, *Chchd3/6* C males = 112. Each week, flies were transferred using a funnel to a clean longer tube with no food and these tubes were placed in a Styrofoam cutout to hold the tubes for tapping. The vial holder was tapped down multiple times until flies were at the base of the vial and then left to record the percentage of flies which reached 10 cm after 10 s. The vial holder was tapped down multiple time to achieve biological replicates = 4 and different batches of genotypes were examined for technical replicates = 5.

### Statistical analyses

All statistical analyses were performed using GraphPad Prism version 8.0.1 for Windows, GraphPad Software, San Diego, California USA, https://www.graphpad.com. Statistical tests used are stated in figure legends. T-tests were performed on most heart assays where only one variable was defined. For tdtK analyses, a ranked one-way ANOVA Kruskal–Wallis test was used. Combinatorial KD assays with

MICOS subunits or mitochondrial fission-fusion genes and *Chchd3/6* loss-of-function in *Drosophila* were analyzed with two-way ANOVA. The interaction plots of two-way ANOVA in *Figure 4—figure supplement 1* were obtained using the "Plot2WayANOVA" function in the "CGPfunctions" package in R.

## PCR
The *Chchd*[D1] line was confirmed via PCR using primers Chchd[D1] F4: ATATATCCGACGATGTGG and Chchd[D1] R4: AGCTCCTGGTTCATCTGG (Q5 High-Fidelity 2 X Master Mix New England Bio).

## Quantitative real time polymerase chain reaction (qRT-PCR)
RNA was extracted using Qiagen miRNeasy Mini Kit and cDNA was synthesized with Qiagen Quanti-Tech reverse transcription kit. qRT-PCR analysis was performed using Roche FastStart Essential DNA Probes Master and Roche LightCycler 96 with 2 biological replicates and 3 technical replicates. Data was analyzed in the LightCycler application. Primers include C*hchd*3/6 F: GCTAGAGGAACTTCAA AGATGG, C*hchd*3/6 R: GGGATAGGAGGATACTTTCGG, *RP49* F: GCTAAGCTGTCGCACAAATG, *RP49* R: GTTCGATCCGTAACCGATGT.

## Human iPSC-derived cardiomyocyte proliferation assays
At day 25 of differentiation, human iPSC-derived cardiomyocytes (hiPSC-CMs) were dissociated with TrypLE Select 10 X (Gibco) for up to 12 min and action of TrypLE was neutralized with RPMI supplemented with 10% FBS. Cells were resuspended in RPMI with 2% KOSR (Gibco) and 2% B27 50 X with vitamin A (Life Technologies) supplemented with 2 µM Thiazovivin and plated at a density of 5000 cells per well in a Matrigel-coated 384-well plate. hiPSC-CMs were then transfected with siRNA (Dharmacon) directed against each gene using lipofectamine RNAi Max (Thermo Fisher). Each siRNA was tested in quadruplicate. Forty-eight hours post-transfection, cells were labeled with 10 µM EdU (Thermo Fisher). After 24 hr of EdU incubation, cells were fixed with 4% paraformaldehyde for 30 min. EdU was detected according to the protocol and cells were stained with cardiac specific marker ACTN2 (Sigma A7811, dilution 1:800) and DAPI. Cells were imaged with ImageXpress Micro XLS microscope (Molecular Devices) and custom algorithms were used to quantify EdU +hiPSC CMs. Cell lines were checked for mycoplasma contamination and genotypic authenticity.

## ATP Measurements
Measurements of ATP were performed using a luciferase assay as described previously (*Liu and Lu, 2010*; *Zanon et al., 2017*). 10–12 hearts per sample were collected from 1-week-old flies and homogenized in 100 µl extraction buffer (100 mM Tris and 4 mM EDTA, pH 7.8) containing 6 M guanidine-HCl followed by rapid freezing in liquid nitrogen. The samples were boiled for 5 min and cleared by centrifugation at 14,000 x *g*. Supernatants were diluted 1:50 and ATP levels were determined using ENLITEN ATP Assay System (Promega, Cat #FF2000) as per manufacturer instructions. Total protein levels were determined by BCA method (Pierce, Cat #23225). ATP measurements were normalized to protein.

## Hybridization chain reaction (HCR)
Hearts were exposed as described above and RNA in situ performed and analyzed as described in *Kirkland et al., 2021*. Briefly, hearts were relaxed using 10 mM EGTA in artificial hemolymph and fixed in 4% formaldehyde in 0.1% Tween 20-PBS for 20 min. Hearts were then washed with 0.1% Tween 20, PBS, 2x5 min. On ice, hearts were incubated in a methanol gradient with PBS for 5 min each (25%, 50%, 75%, 100%, 75%, 50%, 25%). Hearts were then permeabilized in 1% Triton 100 X in PBS for 2 hr at room temperature. The hearts were post-fixed with 4% formaldehyde in 0.1% Tween 20-PBS for 20 min at room temperature before washing on ice with 0.1% Tween 20-PBS, 2x5 min. Subsequently, samples were washed with 50%–0.1% Tween 20-PBS and 50% 5XSSCT (5 X SSC, 0.1% Tween 20, H$_2$O) for 5 min on ice, followed by 5 X SSCT for another 5 min. The hearts were then transferred to a 96-well plate and the hearts incubated in probe hybridization buffer (Molecular Instruments) for 5 min on ice, then 30 min at 37 °C. The solution was then replaced with 2 µl of each probe in 200 µl of probe hybridization buffer and incubated at 37 °C overnight (up to 16 hr). Next, 4x15 min washes were performed with probe wash buffer (Molecular Instruments) at 37° C, then 2x5 min 5XSSCT and

1x5 min amplification buffer (Molecular Instruments). Two µl of corresponding h1 and h2 hairpins (Molecular Instruments) were heated to 95 °C for 90 s, cooled in the dark for 30 min and added to 100 µl of amplification buffer. Hairpin solution was then incubated with the heart samples at room temperature overnight (up to 16 hr), in the dark. Next, samples were washed 2x5 min with 5 X SSCT; 2x30 min with 5 X SSCT; 1x5 min with 5 X SSCT and rinsed 3 x with PBS. DAPI in PBS (1:500) was incubated with the samples for 15 min, then samples were again rinsed 3x5 min in PBS. Samples were then mounted and imaged as described above. To quantify expression, a maximum projection image was created in ImageJ from the confocal stack image and binarized. The region around the cardiomyocyte nucleus was traced and the ROI copied to the binary image for particle analysis. Since segmentation was imperfect for transcripts very close together and to account for differences in pocket size, the % area covered by the transcripts was used to assess statistical significance in Prism (GraphPad).

## Acknowledgements

We gratefully acknowledge the patients and families who participated in this study. We thank Marco Tamayo and Bosco Trinh for excellent technical assistance. This work was supported by National Institutes of Health (R01 HL054732 to R.B.). This work was also supported by a grant from the Wanek Foundation at Mayo Clinic in Rochester, M.N., to J.L.T., T.J.N., T.M.O., R.B. and A.R.C.; by the American Heart Association: AHA Predoctoral Fellowship (18PRE33960593 to K.B.) and AHA Postdoctoral Fellowship (20POST35180048 to N.J.K.).This work was supported by R01 HL153645, R01 HL148827, R01 HL149992, R01 AG071464 (National Institutes of Health to A.R.C). Sanford Burnham Prebys Shared Resources are supported by an NCI Cancer Center Support Grant (P30 CA030199).

## Additional information

### Funding

| Funder | Grant reference number | Author |
| --- | --- | --- |
| Foundation for the National Institutes of Health | HL054732 | Rolf Bodmer |
| Wanek Foundation | Todd and Karen Wanek Family Program for HLHS | Timothy J Nelson<br>Jeanne L Theis<br>Timothy M Olson<br>Rolf Bodmer<br>Alexandre R Colas |
| American Heart Association | 18PRE33960593 | Katja Birker |
| American Heart Association | 20POST35180048 | Natalie J Kirkland |
| Foundation for the National Institutes of Health | HL153645 | Alexandre R Colas |
| Foundation for the National Institutes of Health | HL148827 | Alexandre R Colas |
| Foundation for the National Institutes of Health | HL149992 | Alexandre R Colas |
| Foundation for the National Institutes of Health | AG071464 | Alexandre R Colas |

The funders had no role in study design, data collection and interpretation, or the decision to submit the work for publication.

## Author contributions

Katja Birker, Data curation, Funding acquisition, Validation, Investigation, Writing - original draft; Shuchao Ge, Formal analysis, Funding acquisition, Validation, Investigation, Visualization, Methodology, Writing - original draft, Writing - review and editing; Natalie J Kirkland, Conceptualization, Data curation, Software, Formal analysis, Funding acquisition, Validation, Investigation, Visualization, Methodology, Writing - original draft, Writing - review and editing; Jeanne L Theis, Conceptualization, Data curation, Software, Formal analysis, Investigation, Methodology, Writing - original draft; James Marchant, Formal analysis, Validation, Investigation, Methodology; Zachary C Fogarty, Data curation, Software, Formal analysis, Investigation; Maria A Missinato, Conceptualization, Formal analysis, Validation, Investigation; Sreehari Kalvakuri, Data curation, Formal analysis, Supervision, Funding acquisition, Investigation; Paul Grossfeld, Conceptualization, Formal analysis, Supervision, Writing - original draft; Adam J Engler, Resources, Data curation, Supervision, Funding acquisition, Project administration; Karen Ocorr, Conceptualization, Resources, Data curation, Formal analysis, Supervision, Funding acquisition, Validation, Investigation, Methodology, Writing - original draft; Timothy J Nelson, Conceptualization, Resources, Data curation, Formal analysis, Supervision, Project administration; Alexandre R Colas, Resources, Data curation, Software, Supervision, Funding acquisition, Validation, Investigation, Methodology, Writing - review and editing; Timothy M Olson, Conceptualization, Resources, Formal analysis, Supervision, Writing - original draft, Project administration, Writing - review and editing; Georg Vogler, Software, Formal analysis, Supervision, Validation, Investigation, Visualization, Writing - review and editing; Rolf Bodmer, Conceptualization, Resources, Formal analysis, Supervision, Investigation, Methodology, Writing - original draft, Project administration, Writing - review and editing

## Author ORCIDs

Shuchao Ge ⓘ http://orcid.org/0000-0002-6590-9050
Jeanne L Theis ⓘ http://orcid.org/0000-0002-4494-8683
Zachary C Fogarty ⓘ http://orcid.org/0000-0001-5588-3216
Maria A Missinato ⓘ http://orcid.org/0000-0001-9055-758X
Karen Ocorr ⓘ http://orcid.org/0000-0003-2593-0119
Timothy J Nelson ⓘ http://orcid.org/0000-0002-3862-7023
Alexandre R Colas ⓘ http://orcid.org/0000-0001-8489-0570
Timothy M Olson ⓘ http://orcid.org/0000-0003-2716-9423
Georg Vogler ⓘ http://orcid.org/0000-0002-8303-3531
Rolf Bodmer ⓘ http://orcid.org/0000-0001-9087-1210

## Ethics

Human subjects: Written informed consent was obtained for the index family and an HLHS cohort, under a research protocol approved by the Mayo Clinic Institutional Review Board ("Genetic Investigations in Hypoplastic Left Heart Syndrome", IRB #11-000114). Participants consented to providing clinical health record data, sample procurement for DNA analyses, and publication of de-identified research findings.

## Decision letter and Author response

Decision letter https://doi.org/10.7554/eLife.83385.sa1
Author response https://doi.org/10.7554/eLife.83385.sa2

---

# Additional files

## Supplementary files

• MDAR checklist

• Supplementary file 1. Prioritized candidate genes from HLHS proband. Nine candidate genes harbored rare homozygous variants predicted to impact protein structure or gene regulation. Hom = homozygous, Het = heterozygous, WT = wildtype, gnomAD = Genome Aggregation Database, MAF = minor allele frequency. rsID: reference single nucleotide polymorphism ID. RegulomeDB variants range from 1 (high functional evidence) to 6 (least functional evidence). CADD score (Combined Annotation-Dependent Depletion), higher percentile predicts higher possibility of functionality or pathogenicity.

• Supplementary file 2. Mitochondrial gene screen in the *Drosophila* heart. RNAi lines of different mitochondrial functional groups were selected using the FlyBase.org gene ontology

(GO) term mitochondrion (GO:0005739). The RNAi lines were crossed to Hand[4.2]-Gal4, tdtK and their progeny were assessed for contractility defects at 1-week of adult age using Hand[4.2]-Gal4; tdtK. Note that structural phenotype refers to any visible F-actin phenotype, not specifically to Chchd3/6 KD F-actin phenotype. FS = fractional shortening, DD = End diastolic diameter, SD = systolic diameter.

• Supplementary file 3. CHCHD3 and CHCHD6 MICOS variants in HLHS probands. Rare, predicted-damaging variants in CHCHD3 and CHCHD6 were identified in 6 of 183 HLHS probands in the Mayo Clinic cohort and 1 in the Pediatric Cardiac Genomics Consortium (PCGC). Transcript and relevant protein variants are listed. rsID: reference single nucleotide polymorphism ID. RegulomeDB variants range from 1 (high functional evidence) to 6 (least functional evidence). CADD score (Combined Annotation-Dependent Depletion), higher percentile predicts higher possibility of functionality or pathogenicity. Corresponding *Drosophila* orthologs and orthology scores (DIOPT). *Annotation of variant for a non-canonical transcript (ENST00000448878).

• Supplementary file 4. Candidate genes from HLHS probands harboring CHCHD3 or CHCHD6 variants. The majority of HLHS candidate genes has a *Drosophila* ortholog (genes without ortholog highlighted in green).

## Data availability

Data available on Dryad: https://doi.org/10.5061/dryad.z8w9ghxj1.

The following dataset was generated:

| Author(s) | Year | Dataset title | Dataset URL | Database and Identifier |
|---|---|---|---|---|
| Birker K, Ge S, Kirkland N, Theis J, Marchant J, Fogarty Z, Missinato M, Kalvakuri S, Grossfeld P, Engler A, Ocorr K, Nelson T, Colas A, Olson T, Vogler G, Bodmer R | 2023 | Mitochondrial MICOS complex genes, implicated in hypoplastic left heart syndrome, maintain cardiac contractility and actomyosin integrity | https://dx.doi.org/10.5061/dryad.z8w9ghxj1 | Dryad Digital Repository, 10.5061/dryad.z8w9ghxj1 |

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
