## [Editor Report]

In the revised version, Birker et al. add experiments in flies and human iPSC-derived cardiomyocytes and add new text that further supports their central claim that mutations in MICOS complex components mediate HLHS by reducing sarcomere integrity and heart contractility. The ability to go from identifying variants by whole genome sequencing in HLHS patients to generating oligogenic animal models to test whether these variants produce cardiac phenotypes is well demonstrated here and highlights the importance of model organisms in disease research. Overall, the manuscript is improved, and the data support the claims.

---

## [Decision Letter]

[Editors' note: this paper was reviewed by Review Commons.]

Thank you for submitting your article "Mitochondrial MICOS complex genes, implicated in hypoplastic left heart syndrome, maintain cardiac contractility and actomyosin integrity" for consideration by *eLife*. Your article has been reviewed by 3 peer reviewers at Review Commons, evaluated at *eLife* by 2 reviewers, and overseen by a Reviewing Editor and Detlef Weigel as the Senior Editor.

Based on your manuscript, the reviews, and your responses, we invite you to submit a revised version incorporating the revisions as outlined in your response to the reviews.

When preparing your revisions, please also address the following points:

1) Add new text detailing the nature of the variants and how they impact CHCHD3/6 expression or function.

2) The studies are beautifully done in flies and show clear evidence of loss of F-actin when mitochondrial complex proteins are knocked down. Please confirm that CHCHD3/6 are also involved in regulating cardiomyocyte contractility in the human model such as using human iPSC-derived cardiomyocytes.

3) While the loss of metabolic genes shows significant effects on sarcomere protein expression, there are no data on metabolism other than the immunostaining for the fusion-fission process. It would be helpful to confirm the loss of ATP production or oxygen consumption rates in human iPSC-derived cardiomyocytes that have decreased MICOS complex.

---

## [Author Response]

1. General Statements [optional]

Hypoplastic left heart syndrome (HLHS) is a severe congenital heart disease (CHD) with complex underlying genetics and unclear etiology. Efforts aimed at better understanding the genotype-phenotype relationships of HLHS have yielded an ever-increasing number of candidate genes, but few definitive disease genes, and no satisfying disease mechanism. A major hurdle has been the lack of comprehensive functional evaluation of these CHD/HLHS gene candidates during cardiogenesis in health and disease. Our teams at Sanford-Burnham-Prebys and the Mayo Clinic have joined forces to address this problem, using complementary approaches of genome-wide phenotypic screening, human genetics, and heart functional assays in model systems. This effort identified genes of the MICOS complex as a major candidate pathway to be potentially linked to defects in growth and differentiation in HLHS, thus relevant for elucidating the polygenic nature of HLHS and other CHDs.

2. Description of the revisions based on the planned revisions

Reviewer #1 (Evidence, reproducibility, and clarity (Required)):Major comments:1. The authors mentioned that the heart dysfunction observed upon CHCHD3/6 KD may be mediated via defects in ATP synthase. Then, how does CHCHD3/6 KD affect ATP synthase? Additionally, OPA1 also affects ATP synthase, why does OPA1 KD just reduce fractional shortening (S.T.2) without reducing F-actin staining?

Since MICOS is involved in ETC assembly/sorting in the cristae, ATPase subunits may not be assembled correctly and thus cause defects in mitochondrial morphology and function, which is further supported by reduced mito::GFP staining (see Figure 4B&H) (see discussion, lines 473-483). We reinvestigated F-actin staining upon *Opa1-RNAi-C* KD and indeed found noticeably reduced F-actin staining (see new Supp. Figure 4C), along with reduced fractional shortening (Figure 4L). We thank the reviewer to point this out to give an opportunity to revisit.

To address an involvement of fission-fusion in connection with *CHCHD3/6* KD further, we investigated whether *Opa1*, as well as *Drp1*, interacted with *CHCHD3/6* in affecting contractility and F-actin levels. We conducted the following experiment: *Opa1* and *Drp1* KD and OE in combination with sensitized Chchd3/6-C1 KD (as in Figure 7A,B). We found a significant interaction between *CHCHD3/6* and *Drp1* or *Opa1* co-KD (Figure 4 J,L, Sipp. Figure 3C,I). In addition, *CHCHD3/6* KD and *Opa1* co-KD also exhibited a trend towards a synergistic interaction (Figure 4M, Sipp. Figure 3L), including diminished actin staining. These new results are described under the subheading “Mitochondrial fission-fusion genes, *Drp1* and *Opa1*, interacted genetically with Chchd3/6” (lines 287-315).

2. It has been reported that CHCHD3 KD in HeLa cells causes fragmented mitochondria, so how does CHCHD3/6 KD caused mitochondrial aggregation? What is the mechanism?

Thank you for pointing that out. It is actually more appropriate to call this phenotype “fission-fusion defects” (lines 279, 282, 284, 461).

To address this issue further, we manipulated mitochondrial fission-fusion genes (*Drp1, Opa1*) in conjunction with *Chchd3/6^RNAiC1^*. The results were shown in the new Results section titled “Mitochondrial fission-fusion genes, *Drp1* and *Opa1*, interacted genetically with Chchd3/6”, as summarized above (lines 287-315).

3. The ultrastructure of mitochondria (especially aggregated mitochondria) in control and CHCHD3/6 KD heart of *Drosophila* should be analyzed by TEM.

TEM is difficult to perform since cardiac tissue is extremely thin. It is not clear if fission vs fusion defects can easily be distinguished, even with TEM. Instead, we performed genetic interaction experiments, as summarized in responses to points 1. and 2., above.

Reviewer #1 (Significance (Required)):The manuscript partially illustrates the relationship between MICOS complex with Hypoplastic left heart syndrome (HLHS), which is interesting to the reader.

We thank the reviewer for their appreciation of our study.

Reviewer #2 (Evidence, reproducibility and clarity (Required)):This study performed whole genome sequencing (WGS) on a large cohort of hypoplastic left heart syndrome (HLHS) patients and their families to identify candidate. Nine candidate genes with rare, predicted damaging homozygous variants were identified. Of the candidate HLHS gene homologs tested, cardiac-specific knockdown (KD) of the mitochondrial contact site and cristae organization system (MICOS) complex subunit dCHCHD3/6 resulted in drastically compromised heart contractility, diminished levels of sarcomeric actin and myosin, reduced cardiac ATP levels, and mitochondrial fission-fusion defects. These heart defects were similar to those inflicted by cardiac KD of ATP synthase subunits of the electron transport chain (ETC), consistent with the MICOS complex's role in maintaining cristae morphology and ETC complex assembly. Analysis of 183 genomes of HLHS patient-parent trios revealed five additional HLHS probands with rare, predicted damaging variants in CHCHD3 or CHCHD6. Hypothesizing an oligogenic basis for HLHS, the authors tested 60 additional prioritized candidate genes in these cases for genetic interactions with CHCHD3/6 in sensitized fly hearts. Moderate KD of CHCHD3/6 in combination with Cdk12 (activator of RNA polymerase II), RNF149 (E3 ubiquitin ligase), or SPTBN1 (scaffolding protein) caused synergistic heart defects, suggesting the potential involvement of a diverse set of pathways in HLHS.General Comments:The authors performed an elegant series of experiments that implicate variants of dCHCHD3/6 in HLHS patients as contributing to mitochondrial and sarcomeric defects and contractile function defects. Demonstrating in *Drosophila* the functional and biochemical implications of knocking out dCHCHD3/6 provides some potentially important insights into the functional and biochemical implications of dCHCHD3/6 variants in HLHS patients. The data is also complemented by hiPSC-CM studies in which knockdown of CHCHD6 and CHCHD3 showed similar alterations in ATP synthase and mitochondrial morphology.The authors nicely show that knock down of the subunit dCHCHD3/6 resulted in drastically compromised heart contractility, diminished levels of sarcomeric actin and myosin, reduced cardiac ATP levels, and mitochondrial fission-fusion defects in the Drosphilia.What is not clear is how these changes mirror the phenotype of HLHS in humans. It would be helpful to speculate to a greater extent as to how these changes would manifest as a decreased left ventricular development in HLHS.

This is indeed a very important question we comment on in the discussion (lines 453-460). We want to stress that we focus on genetic interactions in heart development, not on convergent endpoint phenotypes between flies and humans. However, our studies do support the idea that mitochondrial defects could contribute to HLHS. We show that MICOS deficiency causes mitochondrial defects manifest in diminished ATP production in addition to diminished sarcomeric actin and myosin causing diminished contractility. Impaired contractility during development has previously been proposed to contribute to defective human cardiac growth (no flow – no growth, Goldberg and Rychik, 2016; Grossfeld *et al.*, 2019), thereby compounding the potentially polygenic effects from damaging gene variants.

Why there would a preferential effect on the left ventricle is another interesting question. We speculate that some of the patient-specific variants, in addition to MICOS genes, are in genes preferentially affecting the left ventricle, thus preferentially affecting its growth as well as its contractile ability, then again compounded by impaired blood flow feeding back to diminishing growth.

Specific Comments:Line 139: Figure 1A does not show echos from the siblings.

We apologize that the “(Figure 1A)” was in wrong position (after echocardiograms), causing confusion. We moved it to the previous sentence (line 139). In case the reviewers require that echocardiograms are shown as supplemental data, we can provide these.

Line 155: This table is listed as "Table 1" not Supplemental Table 1.

We apologize for mislabeling. This table is now listed as Supplementary Table 1.

Reviewer #2 (Significance (Required)):This is a highly significant study. The main audience would be pediatric cardiologists and geneticists.

We thank the reviewer for their appreciation of our study.

[Editors' note: further revisions were suggested prior to acceptance, as described below.]

Based on your manuscript, the reviews, and your responses, we invite you to submit a revised version incorporating the revisions as outlined in your response to the reviews.When preparing your revisions, please also address the following points:1) Add new text detailing the nature of the variants and how they impact CHCHD3/6 expression or function.

We added the following to the manuscript (lines 394-399): “In total, there were four noncoding variants with a RegulomeDB rank of 2a (n=2), 2b (n=1) and 4 (n=1) providing evidence for haploinsufficiency due to predicted disruption of transcription factor binding in the presence of the variant. The remaining two missense variants, one inherited and one de novo, may alter the protein structure or function and lead to downstream functional consequences.”

2) The studies are beautifully done in flies and show clear evidence of loss of F-actin when mitochondrial complex proteins are knocked down. Please confirm that CHCHD3/6 are also involved in regulating cardiomyocyte contractility in the human model such as using human iPSC-derived cardiomyocytes.

We thank the editor for appreciating the evidence of loss of F-actin on contractility in flies. Like our findings in flies we detect reduced sarcomeric F-actin patterning also in hiPSC-CM (Supp. Figure 7).

3) While the loss of metabolic genes shows significant effects on sarcomere protein expression, there are no data on metabolism other than the immunostaining for the fusion-fission process. It would be helpful to confirm the loss of ATP production or oxygen consumption rates in human iPSC-derived cardiomyocytes that have decreased MICOS complex.

We measured oxygen consumption rate (OCR) in human cardiomyocytes derived from iPSC and found a decrease at 60 minutes after oligomycin treatment in iPSC-CMs with co-KD of *CHCHD3* and *CHCHD6*. See lines 382-386 for the newly added results and Supplementary Figure 7.